



**Seasonal characteristics of atmospheric formaldehyde (HCHO) in a coastal city of**
**southeast China: Formation mechanism and photochemical effects**
Taotao Liu[1,2,3], Yiling Lin[1,4], Jinsheng Chen[1,2]*, Gaojie Chen[1,2,3], Chen Yang[1,2,3], Lingling Xu[1,2], Mengren Li[1,2], Xiaolong
Fan[1,2], Yanting Chen[1,2], Liqian Yin[1,2], Yuping Chen[1,2,3], Xiaoting Ji[1,2,3], Ziyi Lin[1,2,3], Fuwang Zhang[5], Hong Wang[6], Youwei
Hong[1,2]*
[1]Center for Excellence in Regional Atmospheric Environment, Institute of Urban Environment, Chinese Academy of Sciences,
Xiamen, China
[2]Key Lab of Urban Environment and Health, Institute of Urban Environment, Chinese Academy of Sciences, Xiamen, China
[3]University of Chinese Academy of Sciences, Beijing, China
[4]College of Chemical Engineering, Huaqiao University, Xiamen, China
[5]Environmental Monitoring Center of Fujian, Fuzhou, China
[6] Fujian Key Laboratory of Severe Weather, Fujian Meteorological Science Institute, Fuzhou, China
Corresponding authors E-mail: Jinsheng Chen (jschen@iue.ac.cn); Youwei Hong (ywhong@iue.ac.cn)
**Abstract:**
Formaldehyde (HCHO) is a vital reactive carbonyl compound, which plays a critical role in the
atmospheric oxidation capacity (AOC), radical chemistry, and $O_3$ formation. Yet, the majority of the current
studies on HCHO photochemical mechanism in coastal areas remain scarce, thus limiting the full
understanding of potential atmospheric impacts with limited influence from marine sources. Here, field
campaigns were conducted at a typical urban site in southeast China to reveal the characteristics and potential
source of ambient HCHO, as well as its impact on photochemistry, during spring and autumn of 2021. The
result showed that the HCHO mixing ratios were 2.94±1.28 ppbv and 3.19±1.41 ppbv in spring and autumn,
respectively. Secondary formation made the largest contributions to HCHO (49% in spring and 46% in
autumn), followed by vehicle exhaust (25% and 20%) and biogenic emission (18% and 24%) in this study.
Furthermore, in order to identity the impact of HCHO on photochemistry process, the formation pathways and
key precursors (alkenes and aromatics) of secondary HCHO were furtherly investigated based on Observation-
Based Model (OBM). The net HCHO production rate in autumn (−0.40±0.70 ppbv h$^{-1}$) was lower than that
in spring (0.10±0.37 ppbv h$^{-1}$), due to the increase in HCHO loss rate under the intense solar radiation and
relatively low precursor levels to limited HCHO secondary formation. Disabling HCHO mechanism decreased
the abundance of OH (25% in spring and 16% in autumn), $HO_2$ (45%, 40%), and $RO_2$ (26%, 19%). Meanwhile,
the net $O_3$ production rates dropped by 32% in spring and 29% in autumn, which were mainly dominated by
the reduction of radical propagation efficiencies. The analysis of HCHO potential sources, formation pathways,
and impacts on $O_3$ formation provided significant insights into photochemical mechanisms and pollution
control in coastal areas.

**Keywords**: HCHO; Sources apportionment; Formation mechanism; Atmospheric oxidation capacity; Radical
chemistry





## 1 Introduction

Formaldehyde (HCHO) is an important pollutant of photochemistry, and also is one of the most abundant reactive carbonyl compounds in the troposphere, which plays a critical role in the atmospheric oxidation capacity and radical chemistry (Blas et al., 2019; Edwards et al., 2014; Bao et al., 2022). Current studies had found that the production rate of HCHO to hydroxyl radical (OH) and hydroperoxyl radical ($HO_2$) was 1 order of magnitude higher than that of ozone ($O_3$) (Zhang et al., 2021a; Liu et al., 2022a, 2022b). HCHO is also the major precursor of $O_3$ and secondary organic aerosols (Zeng et al., 2019; Possanzini et al., 2002; Liu et al., 2022b). Due to its sensitization, irritation, and mutagenicity, the World Health Organization (WHO) listed HCHO as a dangerous carcinogen (WHO, 2000). In recent years, HCHO has become a research focus because of its important effects on atmospheric chemistry and human health (Zeng et al., 2019).

HCHO is directly emitted from anthropogenic activities (such as vehicle exhausts and industrial activities) and natural emissions (such as biomass burning, vegetation, and sea water) (Luecken et al., 2018; Anderson et al., 2017; Lowe and Schmidt, 1983; Wittrock et al., 2006). Secondary formation of HCHO from the photo-oxidation of volatile organic compounds (VOCs) is also a significant source (Anderson et al., 2017, Zeng et al., 2019). The chemical reactions of VOCs with $OH/NO_3/O_3$ can produce HCHO, in which the alkoxy radical reactions ($RO+O_2$) have significant contributions to HCHO formation (Yang et al., 2020; Ling et al., 2017). The photolysis and oxidation with OH radicals are the main loss pathways of HCHO, which can directly produce $HO_2$ radicals and indirectly produce OH radicals by oxidizing NO to $NO_2$ (Zhang et al., 2021a; Liu et al., 2015). The OH radical is the principal oxidant for atmospheric oxidation capacity (AOC), and efficient ROx recycling (i.e. OH$\rightarrow$$RO_2$$\rightarrow$RO$\rightarrow$$HO_2$$\rightarrow$OH, ROx=OH+$HO_2$+$RO_2$) can produce $O_3$ (Zhang et al., 2021b). Totally, HCHO can modulate $O_3$ formation and AOC levels by radical chemistry, but the influencing mechanisms of HCHO on photochemistry are still complex and unclear, which helps to provide scientific guidance for further control of air pollution.

With the aggravation of $O_3$ pollution, the researches of HCHO have been widely reported around the world because of its significant impact on $O_3$ formation (Li et al., 2014; He et al., 2020; Villanueva et al., 2021; Nussbaumer et al., 2021). However, few studies on systematic field measurement of HCHO were reported in coastal cities with relatively clean atmospheric environment. In China, the relevant studies and observations were mainly focused on the megacities and regions with rapid economic development, such as Beijing, the Yangtze River Delta (YRD), and the Pearl River Delta (PRD) region. These studies about HCHO mainly focused on the pollution characteristics, sources, and identification of the dominant precursors. The major HCHO source is the methane ($CH_4$) oxidation in both regional background/remote PRD regions and suburban



YRD regions, while isoprene ($C_5H_8$) was an important precursor of HCHO in a rural PRD region (Yang et al.,
2021; Yang et al., 2020; Li et al., 2014). For some studies in urban sites of Beijing and the YRD region, alkenes
degradation contributed most to HCHO formation (Ling et al., 2017; Liu et al., 2015). Few studies assessed
that the HCHO photochemical reactions accounted for 9%–14% of atmospheric oxidation and 15% of $HO_2$
formation, and reducing HCHO led to a decrease of 31% in $O_3$ formation (Zhang et al, 2021a; Zeng et al,
2019). Currently, the researches on the influence of HCHO on atmospheric oxidation and photochemistry are
still scarce. Different types and sources for HCHO precursors lead to complicated secondary formation
mechanisms in various regions, thus the exploration of HCHO sources and photochemical effects are very
necessary for ozone pollution mitigation by efficient control strategies.
Xiamen, a coastal city of Southeast China with relatively low atmospheric particles, frequently appeared
$O_3$ pollution events in spring and autumn, when the meteorological conditions were governed by weather
systems such as the quasi-stationary front and the west pacific subtropical high (Liu et al., 2022a; Wu et al.,
2019). The favorable photochemical reaction conditions (including high air temperature, low relative humidity,
intense solar radiation, and stagnant atmosphere) provided is a good 'laboratory' to further explore HCHO
formation mechanism and its impact on $O_3$ formation. In this study, the methods of the Observation-Based
Model with the Master Chemical Mechanism (OBM-MCM) and Positive Matrix Factorization (PMF) model
were employed to better understand the distribution and photochemical behavior of HCHO. Our study aims
to reveal (1) the seasonal characteristics and source apportionment of HCHO, (2) the HCHO formation
mechanism and sensitivity to precursors, and (3) the impacts of HCHO on atmospheric oxidation capacity
(AOC), radical chemistry, and $O_3$ formation.

**2. Methodology**
**2.1 Site descriptions and field measurement**
Xiamen is a typical southeastern coastal city located on the west coast of the Taiwan Strait. Figure 1
showed the location of the observation site (Liu et al., 2022a, 2022b). The observations of multi-parameters
were based on the Atmospheric Environment Observation Supersite (AEOS, 24.61° N, 118.06° E), which was
about 70 m above the ground in the Institute of Urban Environment, Chinese Academy of Sciences in Xiamen.
The site is a typical urban site, surrounded by highways, shopping malls, educational institutions,
administrative, and residential areas. The field campaigns were continuously conducted from May 15 to June
9, 2021, and September 5 to 30, 2021, when the typical photochemical pollution occurred frequently under
the influence of various weather systems.

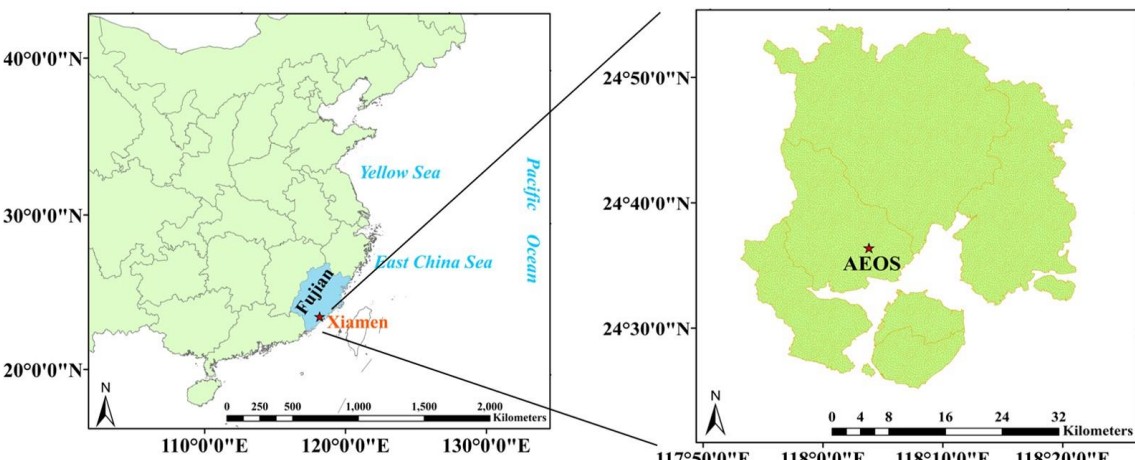

**Figure 1. Location of Xiamen and the observation site.**

HCHO analyzer (FMS-100, Focused Photonics Inc., Hangzhou, China) was used to monitor the HCHO
mixing ratios with a temporal resolution of 1 s, which collected gaseous HCHO by an $H_2SO_4$ stripping solution
and quantified HCHO mixing ratios through detection by fluorescence. The dilutions of the HCHO standard
solution were used to make a multi-point calibration every week for obtaining a curve with $R^2 \geq 0.99$. The
limit of detection was 50 pptv and the uncertainty was ≤5% in this study. A gas chromatography coupled with
a mass spectrometer (GC-FID/MS, TH-300B, Wuhan Tianhong Instruments Co., China) analyzed the VOCs
with a 1-hour time resolution. The flame ionization detector (FID) using a PLOT ($Al_2O_3$/KCl) column (15 m
× 0.32 mm × 6.0 μm) measured the hydrocarbons with 2~5 carbons; a DB-624 column (60 m × 0.25 mm ×
1.4 μm) was used to quantify the other VOCs compounds. The instrument system can quantitatively analyze
106 VOCs in the ambient atmosphere, including 29 alkanes, 11 alkenes, one alkyne, 17 aromatics, 35
halogenated hydrocarbons, and 13 OVOCs (Table S1). The single-point calibration and multi-point calibration
were performed every day and every month with the standard mixtures of PAMS and TO15, respectively. The
detection limits of the VOCs ranged from 0.02 to 0.30 ppbv, and the precision was ≤10%.
PAN analyzer (PANs-1000, Focused Photonics Inc., Hangzhou, China) through gas chromatography with
electron capture detector (GC-ECD) analyzed PAN, and the single-point calibration and the multi-point
calibration were conducted every week and every month, respectively. The precision and accuracy of PAN
measurements were 3% and ±10%, respectively. HONO was monitored by the Monitoring Aerosols and Gases
in Ambient Air (MARGA, ADI 2080, Applikon Analytical B.V., the Netherlands), the uncertainty of which
was ±10%. Criteria air pollutants (i.e. $O_3$, $NO_x$, and CO) were measured by the Thermo Instruments TEI 49i,
42i, and 48i (Thermo Fisher Scientific, Waltham, MA, USA), respectively. The meteorological parameters,





such as pressure (P), air temperature (T), relative humidity (RH), wind speed (WS), and wind direction (WD),
were offered by a weather station with sonic anemometer (150WX, Airmar, USA). Photolysis frequencies (i.e.
$J$HCHO, $J$O$^1$D, $J$NO$_2$, $J$HONO, $J$H$_2$O$_2$, and $J$NO$_3$) were monitored by a photolysis spectrometer (PFS-100,
Focused Photonics Inc., Hangzhou, China). Table S2 shows the detailed uncertainty, detection limit, and time
resolution of instruments for trace gas observation. Strict quality control and quality assurance were applied
to ensure the data validity in our study, and the detailed introductions of the monitoring procedure were
discussed in our previous studies (Hu et al., 2022; Liu et al., 2022a, 2020b).

**2.2 Positive Matrix Factorization (PMF) model**
A Positive Matrix Factorization model (PMF 5.0) was used to identify the sources of HCHO. The
model decomposes a speculated sample matrix into factor contributions and profiles, as shown in
Equation 1 (Norris et al., 2014):
$$e_{ij} = x_{ij} - \sum_{k=1}^{p} g_{ik} f_{kj} \qquad (1)$$
where $e_{ij}$ represents the residual matrix for $j$ species in i sample, $x_{ij}$ is the measured concentration
matrix of $j$ species in $i$ sample, $g_{ik}$ is the factor contribution matrix of $k$ source in $i$ sample, and $f_{kj}$ is the
factor fraction matrix of $j$ species in $k$ source. The $Q(E)$, the model criteria, could evaluate the stability
of the solution and be calculated as follows (Sarkar et al., 2017):
$$Q(E) = \sum_{i=1}^{n} \sum_{j=1}^{m} \left(\frac{e_{ij}}{s_{ij}}\right)^2 \qquad (2)$$
where $s_{ij}$ is the standard deviation of $j$ species in i sample, n and m represent the number of samples and
species, respectively. The species inputting to PMF model were mainly treated as important tracers of the
pollution sources. The species uncertainties need put into the model and were calculated as Equation 3, where
$EF$ is the error factor of 10%, and MDL is the minimum detection limit.
$$u_{ij} = \sqrt{(EF \times conc.)^2 + (MDL)^2} \qquad (3)$$
In this study, 4-6 factors were tested, and all runs converged. We selected the lowest Qrobust in each run
of PMF for further examination. The scaled residual needs to be between -3 and 3 for most data points. The
sensitivity of the model parameters was tested with different "Extra modeling uncertainty". Q values decreased
significantly with increasing uncertainty, and a value of 5% was finally chosen. We compared the lowest value
of $Q_{robust}/Q_{expected}$ as the number of factors increased at each step (Brown et al., 2015). Table S3 shows that
$Q_{robust}/Q_{expected}$ decreased from 4 factors to 6 factors (0.62). Displacement of factor elements (DISP) had no
swaps for all factors, indicating that the solution was valid. The bootstrap (BS) of the results with 6 factors
showed 80% mapping for one factor. For 4 factors, one factor for BS had 75% mapping. However, 5 factors



had more than 90% mapping, and ultimately 5 factors were the optimal solution.

**2.3 Observation-Based Model**
As one of the important methods for analyzing atmospheric chemical processes, the Observation-Based
Model (OBM) has broad application potential in deeply exploring atmospheric observation data and
comprehensively understanding the regional atmospheric pollution (Zhang et al., 2021b). About the chemical
mechanism, the OBM incorporating the latest chemical mechanism version of MCM-v3.3.1 (OBM-MCM,
http://mcm.leeds.ac.uk/MCM/, last access: 13 May 2022) was applied to simulate the detailed photochemical
processes and quantify the reaction rates of HCHO mechanism, and the OBM-MCM model introduced 142
VOCs and about 20000 chemical reactions (Jenkin et al., 2003; Saunders et al., 2003). The physical process
of dry deposition and atmospheric dilution within the boundary layer height (varied from 300 m to 1500 m)
was considered in the model (Li et al., 2018; Liu et al., 2022a, 2022b). Therefore, the dry deposition velocity
of some atmospheric reactants showed in Table S4, which avoided continuous accumulation of pollutant
concentrations in the model (Zhang et al., 2003; Xue et al., 2016).
The observed trace gases (i.e., $O_3$, NO, $NO_2$, CO, $SO_2$, VOCs including HCHO, PAN, and HONO),
photolysis rate constants ($J$HCHO, $J$O$^1$D, $J$NO$_2$, $J$H$_2$O$_2$, $J$HONO, and $J$NO$_3$), and meteorological parameters
(i.e., RH, T, and P) with a time resolution of 1 h were put into the OBM-MCM model, which were updated at
1 h intervals in the OBM-MCM model to constraint and localize the model. The other photolysis rates (such
as OVOCs photolysis rates) were parameterized by the measured $J$(NO$_2$) and the solar zenith angle (Saunders
et al., 2003). Before running the model, the model was pre-run for 2 days to constrain the unmeasured species
(e.g., OH, HO$_2$, and RO$_2$ radicals) reaching a steady state (Liu et al., 2022b).
The HCHO can affect $O_3$ formation and atmospheric oxidation capacity by radical chemistry (Yang et
al., 2020; Li et al., 2014), hence the formation and loss of HCHO were discussed in our study. Furthermore,
the HCHO sensitivities to their precursors were analyzed by relative incremental reactivity (RIR) (Eq. 4)
(Chen et al., 2020). P(HCHO) means the net production rate of HCHO, which was calculated by the
differences between HCHO production rate and loss rate (Chen et al., 2014). The ΔX/X represents the
reduction ratio of each targeted HCHO precursor group, and the value adopted is 20% (Liu et al., 2022a,
2022b).
$$\text{RIR(HCHO)} = \frac{\Delta P(HCHO)/P(HCHO)}{\Delta X/X} \qquad (4)$$




**3. Results and discussion**

**3.1 Overview of observations**

The occurrence of ambient HCHO, air pollutants, and meteorological parameters were shown in Fig. 2, and the related statistical information was summarized in Table S5. The average levels of the measured HCHO in spring and autumn were 2.92±0.27 ppbv and 3.16±1.41 ppbv, respectively. The average HCHO level throughout the observation campaign (3.07±1.35 ppbv) in Xiamen was lower than that in megacities, such as Guangzhou (summer: 6.94±3.36 ppbv) (Ling et al., 2017), Shenzhen (summer: 5.00±4.40 ppbv) (Wang et al., 2017), and Beijing (summer: 11.17±5.32 ppbv) (Yang et al., 2018), but were comparable to various coastal cities with relatively clean air, including Fuzhou (Spring: 2.54±2.09 ppbv) (He et al., 2020), Shantou (autumn: 4.12±1.02 ppbv) (Shen et al., 2021), and Hong Kong (spring: 3.36±0.75 ppbv) (Lui et al., 2017), indicating the influence of anthropogenic activities and photooxidation capacity.

The average mixing ratios of HCHO in autumn were 1.08 times higher than those in spring, which was consistent with previous findings in South China (Lui et al., 2017; Wang et al., 2017). There was a relatively favorable photochemical reaction condition in autumn compared with those in spring. The $O_3$ mixing ratios in autumn (36.25±22.36 ppbv) were 1.23 times higher than that in spring (29.52±15.97 ppbv). Correlation analysis among HCHO, air pollutants, and meteorological parameters was shown in Table S6. As a typical secondary photochemical product, $O_3$ had significantly positive correlations with air temperature (0.40 in spring and 0.52 in autumn) and $J$HCHO (0.49 in spring and 0.61 in autumn), indicating that meteorological conditions obviously influenced photochemical reactions in autumn. Also, the correlation between TVOCs and HCHO in autumn (0.54) was more remarkable than that in spring (0.44). Previous studies found that high values of HCHO were mainly caused by the strong photo-oxidation of VOCs (Wolfe et al, 2016). In this study, the mixing ratios of isoprene were higher in autumn (0.41±0.54 ppbv) than that in spring (0.33±0.38 ppbv), and isoprene had a significant correlation with HCHO of 0.33 in spring and 0.64 in autumn. Correlation analysis showed $O_3$ and PAN had non-negligible relationships with HCHO in both seasons, and the scatter plots of HCHO along with $O_3$ and PAN were shown in Fig. S1. High correlations of these secondary products represented the dominance of local photochemistry during the observation period.



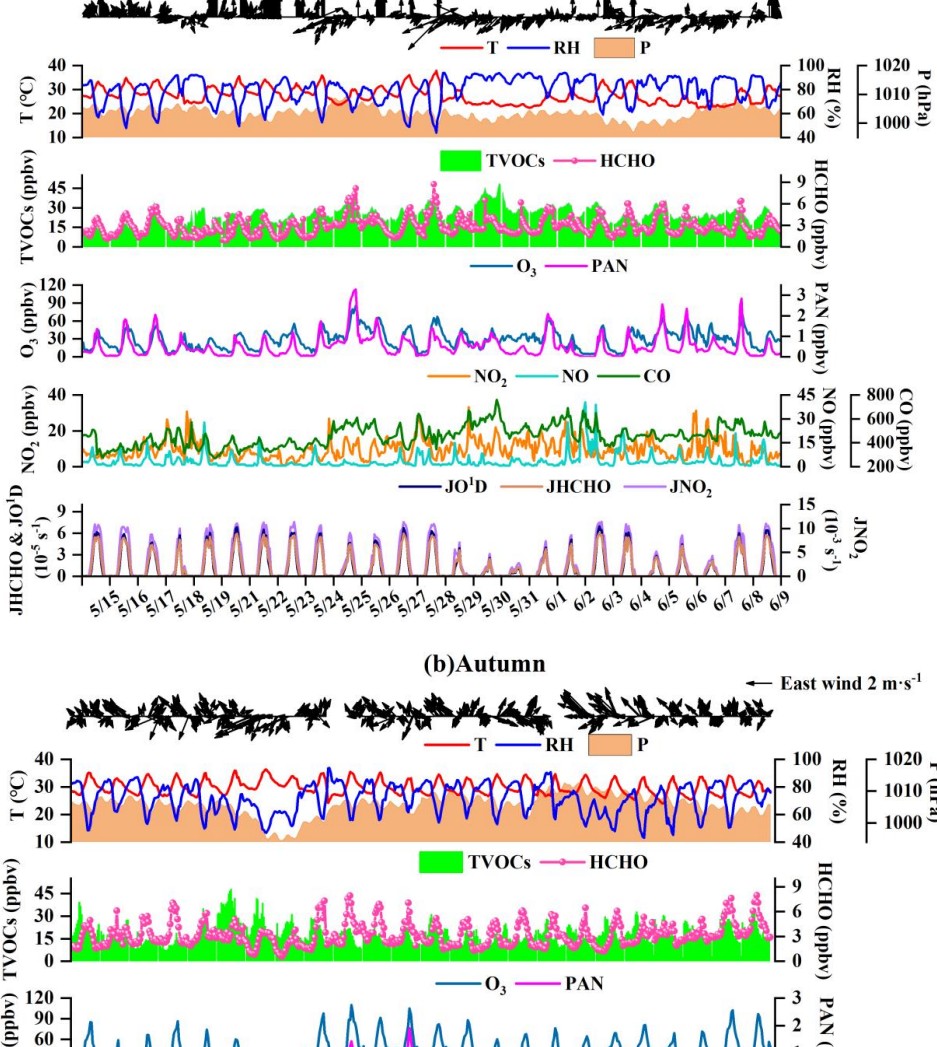


**Figure 2. Time series of HCHO, air pollutants, and meteorological parameters photolysis rate constants**
**in (a) spring and (b) autumn.**





As shown in Fig. 3, the diurnal variations of HCHO presented an increasing trend after sunrise at 06:00
LT (local time), peaked in the afternoon (13:00 LT), and then gradually decreased after sunset. Although the
measured HCHO exhibited similar single peak variations in both seasons, the HCHO mixing ratios kept a
relatively high level during the nighttime (2.34 ppbv in spring and 2.45 ppbv in autumn) compared with those
during the daytime (3.49 ppbv in spring and 3.93 ppbv in autumn), which were attributed to the replenishment
of HCHO primary emissions and accumulation of pollutants under stable weather conditions during the
nighttime. Low wind speed at night (1.13 m·s$^{-1}$ in spring and 1.73 m·s$^{-1}$ in autumn) was favorable for the
accumulation of air pollutants. HCHO concentration (2.09 ppbv in spring, 2.20 ppbv in autumn) was relatively
low at nighttime (0:00 LT-5:00 LT), due to the influence of background contributions. While the relatively
high concentration of HCHO (2.59 ppbv in spring, 2.68 ppbv in autumn) was accumulated between 18:00 LT
to 23:00 LT, related to the influence of the primary HCHO emissions (e.g. vehicle exhausts) and the variety
of boundary layer height. In contrast, HCHO has a short lifetime of several hours due to the quick
decomposition through photolysis and reaction with OH radicals during the daytime (Lowe and Schmidt, 1983;
Zhou et al., 2007). Similar diurnal patterns of HCHO, PAN, and $O_3$ verified the significant effects of local
photochemical formation (Wang et al., 2017; Blas et al., 2019), which was consistent with the findings in
previous studies (Lui et al., 2017; Ling et al., 2017; Zhang et al., 2021; Yang et al., 2018). Diurnal variations
of NOx and CO, important indicators of vehicle emissions, showed a peak around 08:00 LT during the rush
hours, and then increase to high values after decreasing to a minimum at 14:00-16:00 LT, which should be
attributed to the impacts of photochemical depletion reaction and the height alteration of the planetary
boundary layer. In addition, meteorological parameters (T, RH, $J$HCHO, and $J$O$^1$D) in autumn have a
significant difference from those in spring.

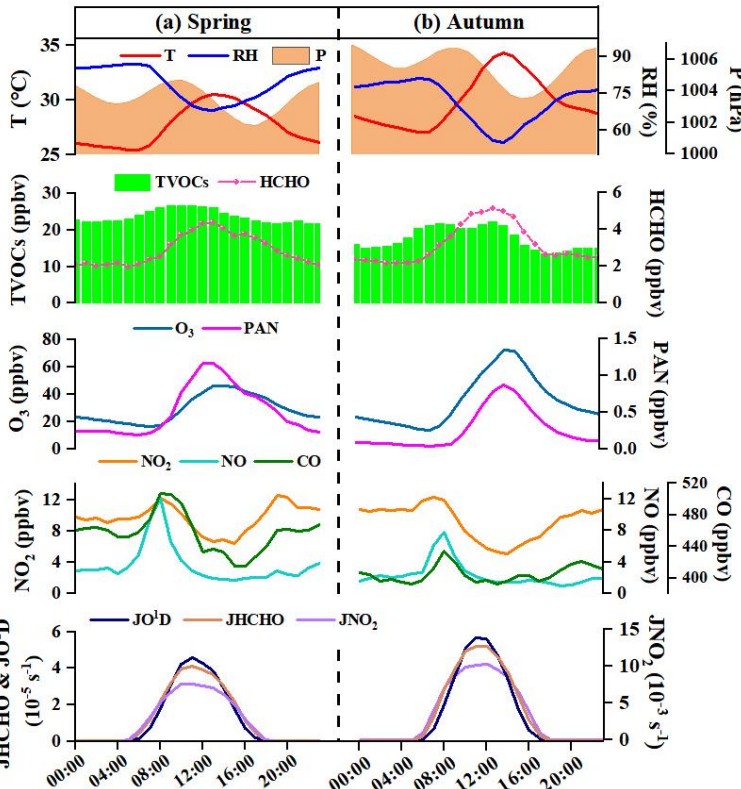


**Figure 3. Average diurnal variations of measured HCHO, air pollutants, meteorological parameters, and photolysis rate constants during (a) spring and (b) autumn.**

**3.2 Source apportionment of HCHO**

PMF was used to analyze the primary and secondary sources of HCHO. As shown in Fig. S2, five factors were identified by the PMF model. Factor 1 was characterized by a high load of $O_3$, which was attributed to intense photochemical processes and secondary formation (Li et al., 2010). Factor 2 has high loadings of 3-methylpentane, iso-pentane, the light hydrocarbons of n/iso-pentane and n/iso-butane, and aromatics. Therefore, factor 2 is defined as the source of vehicle exhaust (Li et al., 2017; Liu et al., 2008). Factor 3 contributed significantly to alkenes and aromatics, such as propene, 1-butene, ethene, and benzene, which were the main VOC species in petrochemical industry (Sinha et al., 2019; Wu et al., 2016; Guven and Olaguer, 2011). This factor was identified as industrial emission. Factor 4 was characterized by a high percentage of isoprene, and was designated as biogenic source (Sindelarova et al., 2022; Na et al., 2004). Factor 5 has high loadings of toluene and 1,2-dichloroethane, which were widely used as industrial solvents and laboratory reagents (Mo et al., 2017). So, factor 5 was identified as solvent usage.

The percentages of different sources to ambient HCHO in spring and autumn was shown in Fig. 4. The





contribution of secondary formation (49% in spring and 46% in autumn) to HCHO was the largest, comparable
to other urban sites such as Guangzhou (53%) (Ling et al., 2017) and Hong Kong (53%) (Lui et al., 2017).
Previous studies have reported that secondary formation was generally the main source of HCHO (34%~70%)
(Guven and Olaguer, 2011; Ling et al., 2017; Wang et al., 2017; Zeng et al., 2019). The contribution of HCHO
from vehicle exhaust in spring (25%) was higher than that in autumn (20%), partly attributed to the
unfavorable diffusion conditions in spring. According to backward trajectories analysis (Fig. S3), air mass
(87%) in spring originated from the southwest, which passed through Xiamen downtown areas with large
amounts of vehicle exhausts emissions. The variation of biogenic source showed a clear seasonal trend, and
contributed 18% in spring and 24% in autumn to the ambient HCHO, which consisted of isoprene levels. The
contribution of biogenic source in Xiamen with high vegetation coverage was higher than that in an urban site
of Wuhan (9%) (Zeng et al., 2019). The industrial source in autumn was 1.5 times higher than that in spring,
which could be attributed to long-range transport from the northeast. Backward trajectories (Fig. S3) in autumn
showed 55% air mass transport from the northeast, which brought pollutants from Quanzhou city, an industrial
city adjacent to Xiamen. The contributions of solvent usage to HCHO seemed to be minor. The results were
similar to those at the urban sites (Zeng et al., 2019). Totally, secondary formation, vehicle exhaust, and
biogenic source made significant contributions to HCHO with total contributions of 48%, 23%, and 21%,
respectively.

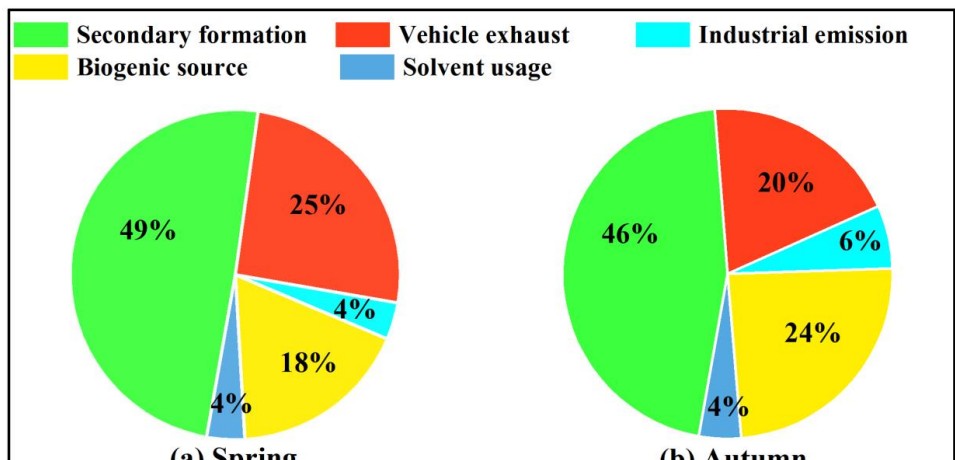

**Figure 4. Seasonal variation of various source contributions to HCHO levels in (a) spring and (a)**
**autumn.**




### 3.3 HCHO formation mechanisms

### 3.3.1 HCHO in situ formation pathways

Figure 5 shows the production and loss pathways of HCHO based on OBM-MCM model. HCHO production rates during the daytime (06:00-17:00 LT) in spring and autumn were 1.89±0.99 ppbv $h^{-1}$ and 1.97±1.16 ppbv $h^{-1}$, while HCHO loss rates were 1.88±1.37 ppbv $h^{-1}$ and 2.47±1.87 ppbv $h^{-1}$, respectively. The results showed that HCHO loss rates in autumn were higher than that in spring. This could be attributed to the favorable meteorological conditions for photolysis, such as intense solar radiation, high air temperature, and low relative humidity, but relatively low levels of air pollutants in autumn compared to those in spring. The daytime average net production rate of HCHO ($NP_{HCHO}$) was 0.10±0.37 ppbv $h^{-1}$ in spring, comparable to that in an island of Hong Kong (0.12 ppbv $h^{-1}$), an urban site (0.18 ppbv $h^{-1}$) and a roadside site (0.16 ppbv $h^{-1}$) of Wuhan, but the $NP_{HCHO}$ (−0.40±0.70 ppbv $h^{-1}$) in autumn was relatively low, due to the increase of HCHO loss rate under the intense solar radiation and relatively low precursor levels to limited HCHO secondary formation (Zhang et al., 2021a; Yang et al., 2020). The $NP_{HCHO}$ reached two peaks at around 07:00 and 16:00 in both seasons (0.47 ppbv $h^{-1}$ in spring and 0.35 ppbv $h^{-1}$ in autumn), and presented the lowest values during 12:00-13:00 LT, when the favorable meteorological conditions made HCHO decomposition more competitive. This result verified that the strong photochemical reaction was conducive to the HCHO production, but also limited the high HCHO value. Previous studies also found that the $NP_{HCHO}$ showed negative values and reached the lowest at noon (Zhang et al., 2021a; Zeng et al., 2019).

The dominant pathway of daytime average HCHO production rate was the $RO+O_2$ reaction, and the reaction rates were 1.68±0.90 ppbv $h^{-1}$ and 1.71±1.02 ppbv $h^{-1}$, which accounted for 87% and 85% of all HCHO production pathways in spring and autumn, respectively. After further refinement of the $RO+O_2$ reactions by classifying different RO first-generation precursors, the $CH_3O+O_2$ pathway contributed $RO+O_2$ reaction rates mostly for 65% and 67%, and contributed to total HCHO production rates of 57% and 58% in spring and autumn, respectively. Moreover, RO derived from alkenes and isoprene reacting with $O_2$ contributed to total HCHO production of 0.41±0.22 ppbv $h^{-1}$ (21%) and 0.11±0.05 ppbv $h^{-1}$ (6%) in spring, and 0.37±0.22 ppbv $h^{-1}$ (18%) and 0.12±0.07 ppbv $h^{-1}$ (6%) in autumn, respectively, and the contributions of RO produced by alkanes and aromatics were less than 2%. The 'others' category (detailed reactions showed in Table S7) accounted for 9% and 11% of the total production rate in spring and autumn, respectively. As for the loss pathways of HCHO, the reaction rates of HCHO+OH were 1.01±0.85 ppbv $h^{-1}$ (55%) and 1.47±1.24 ppbv $h^{-1}$ (61%), and the HCHO photolysis was 0.84±0.53 ppbv $h^{-1}$ (45%) and 0.97±0.63 ppbv $h^{-1}$ (39%) in spring and autumn, respectively. It was worth noting that the contributions of HCHO production pathways





had minor seasonal variations, while the contributions of HCHO loss pathways in autumn were significantly
higher than those in spring. The results implied strong photochemical effects with high yields of ROx radical
in autumn (the detailed description of ROx showed in Fig. 9 of Section 3.4.2).

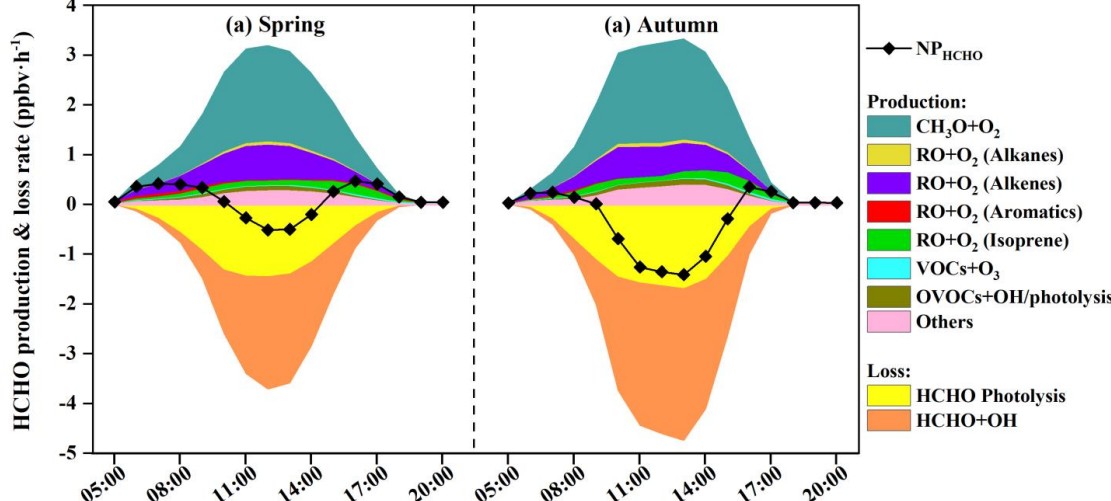


**Figure 5. Model-simulated in situ HCHO production rate and loss rate in (a) spring and (b) autumn.**

**3.3.2 Identification of key precursor species of HCHO**
Sensitivity tests based on the OBM-MCM model were furtherly carried out to quantify the potential
influence of different precursors on HCHO formation (Liu et al., 2022a; Yang et al., 2020). Figure 6 shows
the relative incremental reactivity (RIR) for major groups and specific species of HCHO precursors. HCHO
production was highly VOCs-sensitive with positive RIR values, indicating that reducing VOCs emissions
might effectively inhibit the HCHO formation in Xiamen. As shown in Fig. 6(a), the formation of HCHO was
mainly controlled by alkenes with the largest RIR values in spring (1.19) and autumn (1.05), followed by
isoprene (0.58 in spring and 0.78 in autumn), aromatics (0.75 in spring and 0.55 in autumn), and alkanes (0.45
in spring and 0.43 in autumn). The results suggested that both biogenic and anthropogenic emissions
influenced the HCHO secondary formation. In addition, the RIR of isoprene in spring was lower than that in
autumn, which was attributed to the seasonal characteristics of isoprene concentrations affected by solar
radiation and air temperature (Blas et al., 2019).
Based on the explicit mechanism in the OBM-MCM model, HCHO precursors at the species level could
be further identified. In this study, the impact of anthropogenic precursors on HCHO was mainly discussed.
Alkenes and aromatics made the greatest contributions to HCHO formation among the top 10 VOCs species





(Fig. 6(b)). The top 5 species of RIR were propene (0.35 in spring and 0.31 in autumn), ethene (0.35 and 0.18),
toluene (0.27 and 0.17), m/p-xylene (0.21 and 0.10), and trans-2-butene (0.15 and 0.15), related to their
concentrations in different seasons (Table S5). In addition, for the contributions of anthropogenic emissions
to HCHO formation, vehicle exhaust and biomass burning were the dominant contributors, followed by
solvent usage (Ling et al., 2017; Sinha et al., 2019). These results were consistent with HCHO source
apportionment of PMF in Section 3.2, and also with previous studies in Wuhan and Hong Kong (Zeng et al.,
2019; Yang et al., 2020).

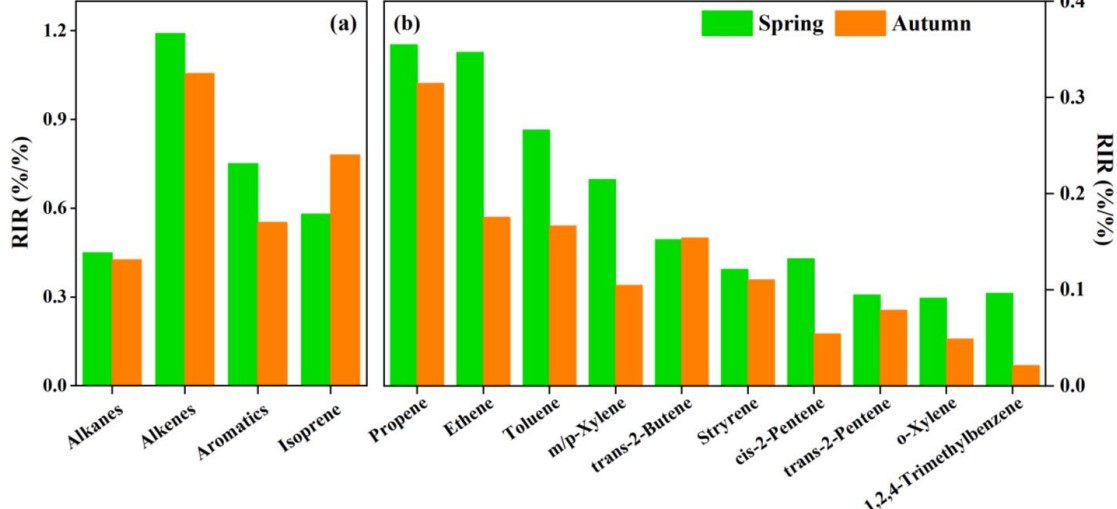


**Figure 6. The OBM-MCM calculated relative incremental reactivity (RIR) for (a) major HCHO**
**precursor groups and (b) top 10 specific species in spring and autumn during the daytime (06:00-17:00**
**LT).**

**3.4. Contribution of HCHO to atmospheric photochemistry**
**3.4.1 Impacts on atmospheric oxidation**
The atmospheric oxidation capacity (AOC) is a crucial aspect of exploring the complex atmospheric
photochemistry processes, reflecting the essential driving force in the loss of primary components and the
production of secondary pollutants in tropospheric chemistry (Chen et al., 2020). AOC was defined as the sum
of oxidation rates in converting primary pollutants (CO, VOCs, etc.) into secondary pollutants by the major
oxidants (i.e., OH, $NO_3$, $O_3$) (Xue et al., 2016). Figure 7 shows the diurnal patterns of the model-calculated
AOC during the observation period. The daily maximum AOC was shown at around 12:00 LT with levels of
$1.24 \times 10^8$ molecules $cm^{-3}$ $s^{-1}$ in spring and $1.48 \times 10^8$ molecules $cm^{-3}$ $s^{-1}$ in autumn, which was comparable to
that in the suburban site of the YRD region ($1.24 \times 10^8$ molecules $cm^{-3}$ $s^{-1}$), higher than that in a regional





background in Hong Kong ($6.2 \times 10^7$ molecules cm$^{-3}$ s$^{-1}$) and a rural site with much low pollution sources in
Berlin ($1.4 \times 10^7$ molecules cm$^{-3}$ s$^{-1}$), but lower than that in some cities, such as Santiago ($3.2 \times 10^8$ molecules
cm$^{-3}$ s$^{-1}$) (Zhang et al., 2021a; Xue et al., 2016; Geyer et al., 2001; Zhu et al., 2020). The AOC levels in
different regions were mainly controlled by the precursors and photochemical conditions, such as solar
radiation and air temperature. As Fig. 7 shows, the OH played a dominant role in contribution to AOC during
the daytime, accounting for around 97% of total AOC, then O$_3$ and NO$_3$ contributed 2% and 3% in both seasons.
During the nighttime, NO$_3$ (71% in spring and 66% in autumn) contributed the most, followed by OH (15%
and 21%) and O$_3$ (14% and 13%). In particular, the AOC by NO$_3$ contributed to the maximum at around 19:00
LT of 84% in spring and 71% in autumn, when relatively high concentrations of O$_3$ and NO$_2$ with weak solar
radiation accelerated the formation and accumulation of NO$_3$ (Fig. 3) (Rollins et al., 2012; Chen et al., 2020).
The AOC levels in autumn were 1.20~1.43 times higher than that in spring, due to the favorable photochemical
conditions. And, the main contribution of AOC was OH radicals, which greatly caused the production of
secondary pollutants.

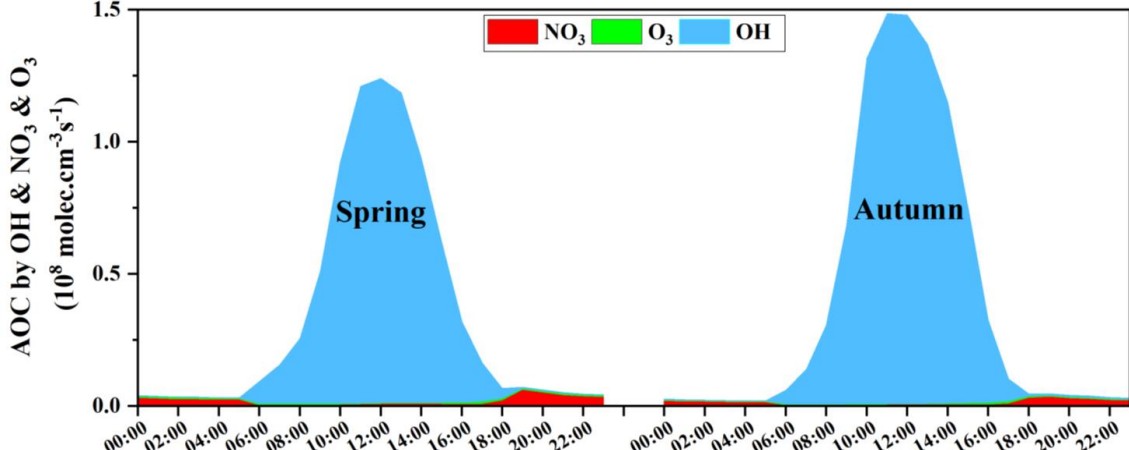

**Figure 7. The model-calculated atmospheric oxidation capacity (AOC) in spring and autumn.**

OH reactivity was used to compare the importance of different reactants to the OH loss, and the model-
calculated OH reactivity in spring and autumn were shown in Fig. 8. The daytime average OH reactivities
were $12.82 \pm 2.73$ s$^{-1}$ in spring and $10.00 \pm 2.30$ s$^{-1}$ in autumn, which were much lower than those in polluted
urban regions, but higher than that in remote or background sites (Lou et al., 2010; Kovacs et al., 2003; Ren
et al., 2005; Zhu et al., 2020). Here, the OH reactivity includes the OH oxidation of both measured species
(such as NOx, CO, C$_2$-C$_{10}$ hydrocarbons, and measured carbonyls) and modeled compounds (mainly including





unmeasured higher OVOCs), and the OH reactivity from the measured compounds accounted for the majority
of the total OH reactivity, thus the OH reactivity from the modeled results might be subject to some uncertainty
due to the lack of direct observations. Compared to measured OH reactivity in other regions of China, the
daily median OH reactivity was $20\pm11$ s$^{-1}$ at an urban site in Beijing and $31\pm20$ s$^{-1}$ at a suburban site in Heshan
of Guangdong Province (Yang et al., 2016, 2017). At a rural site in Wangdu, measured OH reactivity values
ranged between 10 and 20 s$^{-1}$, and the median value during the daytime was 12.4 s$^{-1}$ (Fuchs et al., 2017). The
simulated OH reactivities in our study were within the measured range of other sites in China. Oxygenated
volatile organic compounds (OVOCs, 30% in spring and 31% in autumn), NO$_2$ (27% and 31%), and CO (28%
and 28%) showed large fractions of OH reactivity, followed by NO (6% and 5%), alkenes with relatively high
reactivity compared with hydrocarbons (6% and 4%), alkanes (3% and 3%), and aromatics (3% and 2%). It
should be noted that HCHO accounted for 28% in spring and 34% in autumn of the OH reactivity by OVOCs,
and contributed 8% in spring and 10% in autumn to the total OH reactivity, elucidating the significance of
HCHO in photochemistry. A previous study also showed the importance of HCHO in atmospheric radicals of
HO$_2$ and O$_3$ formation (Zeng et al., 2019).

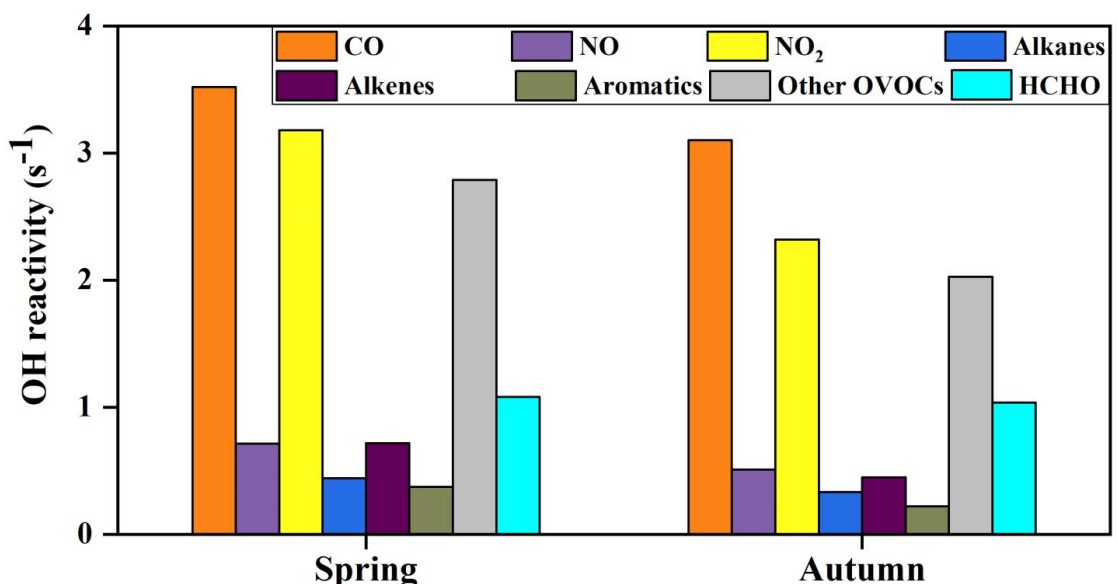


**Figure 8. The model-calculated OH reactivity in spring and autumn during the daytime (06:00-17:00**
**LT).**


### 3.4.2 Impacts on the diurnal patterns of radicals

HCHO modulates $O_3$ formation mainly by controlling the radical recycling in the troposphere (Zeng et al., 2019). To furtherly quantify the changes in ROx chemistry and $O_3$ formation in response to HCHO chemistry, two parallel scenarios were conducted through OBM-MCM model. One scenario was run with all MCM mechanism defined as AS, and the other was run with the HCHO mechanism disabled in MCM mechanism defined as DS. The loss pathways of HCHO (mainly including its photolysis and oxidation with OH radical producing $HO_2$ radical, detailed information in Section 3.3.1) played the key role in the most significant impacts of HCHO on atmospheric photochemistry, thus the HCHO loss pathways were disabled in DS scenario.

Figure 9 shows the diurnal trends of OH, $HO_2$, and $RO_2$ radicals in model scenarios of AS and DS. The ROx showed a significant decline in DS compared to AS. Both the levels and the differences in ROx between AS and DS were higher in autumn than those in spring, which was caused by the intensity of solar radiation (according to the photolysis frequencies in Fig. 9). Anymore, Figure S4 showed the key simulated production and loss rates of ROx, and we found only the ROx loss rate of $OH+NO_2 \rightarrow RO_2$ in spring (2.06 ppbv h$^{-1}$) was significantly higher than that in autumn (1.79 ppbv h$^{-1}$), while the other production and loss rates in spring were lower or comparable than those in autumn. This result could be explained as that high $NO_2$ in spring consumed OH much resulting in low OH level, then low OH concentration suppressed the $RO_2$ production through OH+VOCs reactions leading to less RO production via $RO_2$+NO pathways in spring. These discussions also furtherly testified to the seasonal differences in HCHO+OH reaction rate in Section 3.3.1. For AS, the maximum daily values of OH, $HO_2$, and $RO_2$ concentrations in spring were $1.52 \times 10^7$, $6.17 \times 10^8$, and $3.08 \times 10^8$ molecule·cm$^{-3}$, and those in autumn were $2.35 \times 10^7$, $1.12 \times 10^9$, and $5.09 \times 10^8$ molecule·cm$^{-3}$, respectively. The measured values of ROx in Xiamen were lacking, and we compared the measured values in other regions of China. The maximum daily values of OH and $HO_2$ were in the range of $(4–17) \times 10^6$ molecule cm$^{-3}$ and $(2–24) \times 10^8$ molecule cm$^{-3}$ at both the suburban site and rural site during summer in the North China Plain, respectively (Lu et al., 2012; Tan et al., 2017). The air temperature of Xiamen in autumn was very high and close to that of summer, thus the simulated OH and $HO_2$ concentrations in our study were comparable with the measured results of other places in China. Anymore, previous studies verified that modeled and measured OH agree well when NO mixing ratios were above 1 ppbv, and a continuously increasing underprediction of the observed OH was found towards lower NO concentrations (Lu et al., 2012). The daytime average difference values of OH, $HO_2$, and $RO_2$ between AS and DS in spring were $1.81 \times 10^6$, $1.10 \times 10^8$, and $0.32 \times 10^8$ molecule·cm$^{-3}$, while those in autumn were $1.82 \times 10^6$, $1.89 \times 10^8$, and $0.40 \times 10^8$





molecule·cm$^{-3}$, respectively (Fig. 9). The importance of HCHO in ROx chemistry indicated the necessity to
study the inherent ROx recycling mechanisms.

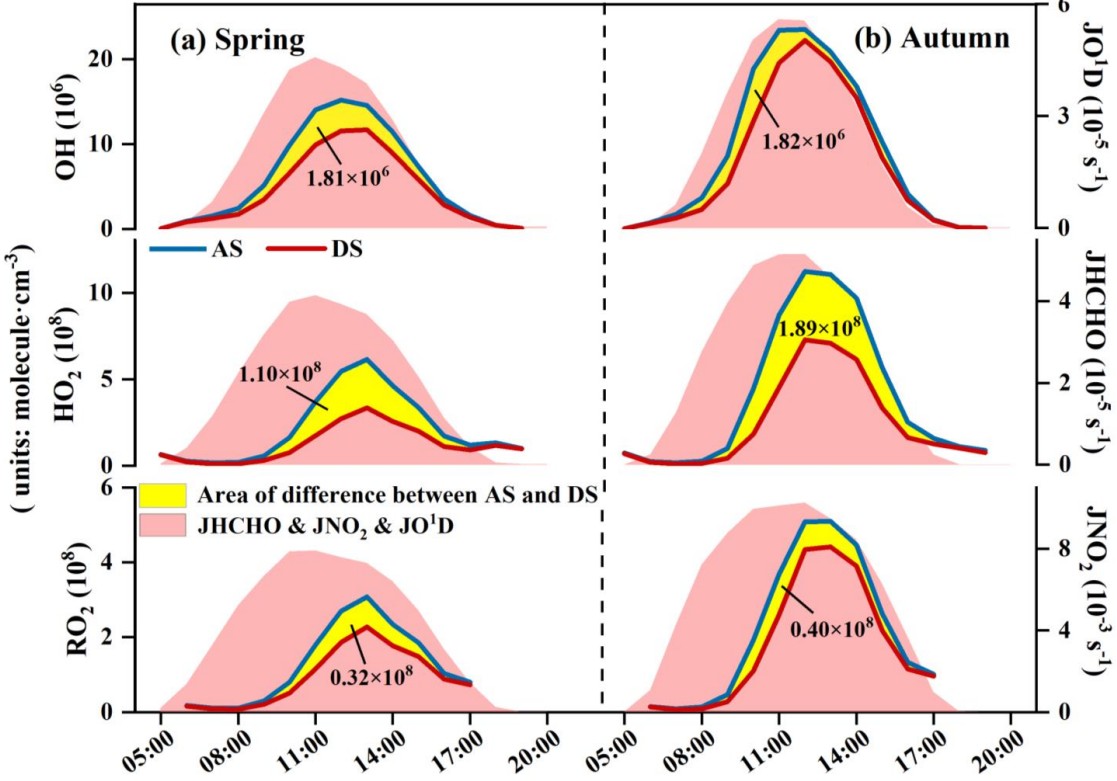


**Figure 9. The diurnal patterns and differences of OH, HO$_2$, and RO$_2$ radicals in model scenarios of AS**
**and DS in (a) spring and (b) autumn. AS scenario was run with all MCM mechanism, and DS scenario**
**was run with the HCHO mechanism disabled in MCM mechanism.**

**3.4.3 Impacts on the formation pathways of radicals**

441        To investigate the chain effect of HCHO on ROx cycling, the formation pathways of radicals were also

analyzed by the OBM model in DS and AS scenarios. Fig. 10 shows the differences in ROx production
pathways in model scenarios of AS and DS. The daytime average OH decreased by 25 % in spring and 16 %
in autumn when HCHO mechanism were disabled. HO$_2$+NO, O$_3$ photolysis, and HONO photolysis were
critical pathways for OH production, which were 7.78, 0.71, 0.53 ppbv h$^{-1}$ in spring and 8.57, 1.06, 0.45 ppbv
h$^{-1}$ in autumn, respectively. The reaction rates of O$_3$+VOCs, HNO$_3$ photolysis, H$_2$O$_2$ photolysis, and OVOCs
photolysis were all below 0.02 ppbv h$^{-1}$, and their changes between AS and DS could be ignored. In Table S8,
disabling HCHO mechanism in DS mainly slowed down the OH production pathway of HO$_2$+NO, which
decreased by 2.75 ppbv h$^{-1}$ (35 %) in spring and 2.61 ppbv h$^{-1}$ (30%). The decrease of O$_3$ photolysis from AS



to DS was 0.06 ppbv h$^{-1}$ (9% in spring) and 0.08 ppbv h$^{-1}$ (8%) in autumn, and the decrease of HONO

photolysis from AS to DS was 0.06 ppbv h$^{-1}$ (12%) in spring and 0.04 ppbv h$^{-1}$ (9%) in autumn. The daytime

average HO$_2$ decreased by 45% in spring and 40% in autumn between AS and DS. Except for the O$_3$+VOCs,

the other HO$_2$ production pathways of OH+CO, RO$_2$+NO, OH+VOCs, HCHO photolysis, and OVOCs

photolysis showed relatively high contributions with production rates of 2.91, 1.77, 1.30, 0.79, 0.38 ppbv h$^{-1}$

in spring and 3.93, 1.82, 1.83, 0.92, 0.36 ppbv h$^{-1}$ in autumn, respectively. Among them, the difference in the

HCHO photolysis between AS and DS was most with 100% reduction of HO$_2$ production, followed by

OH+VOCs (84% in both seasons), OH+CO (26% in spring and 17% in autumn), RO$_2$+NO (11% and 6%),

and OVOCs photolysis (2% and 1%). The daytime average RO$_2$ decreased by 26 % in spring and 19 % in

autumn from AS to DS. Similar to the analysis above, OH+VOCs (2.75 ppbv h$^{-1}$ in spring and 2.74 ppbv h$^{-1}$

in autumn) and OVOCs photolysis (0.33 ppbv h$^{-1}$ and 0.34 ppbv h$^{-1}$) represented the remarkable importance

on RO$_2$ production. The deletion of the HCHO loss reactions also led to a decrease in the daytime reaction

rate of OH+VOCs by 0.53 ppbv h$^{-1}$ (19%) in spring and 0.29 ppbv h$^{-1}$ (10%) in autumn. The decreasing

percentages of OVOCs photolysis were 5% in both seasons, and the changes of NO$_3$+VOCs and O$_3$+VOCs

were slight. The differences in ROx concentration between AS and DS in autumn were higher than that in

spring, but the decrease percentage in autumn was lower than that in spring due to the ROx levels. Meanwhile,

the ROx production rates in autumn were lower than those in spring due to the limited VOCs levels.

HCHO photolysis was the major pathway for HO$_2$ production, thus the differences in HO$_2$ between AS

and DS were the highest, followed by RO$_2$ and OH. Deleting the reactions of HCHO on HO$_2$ production

pathway would decrease the OH production due to the key reaction of HO$_2$+NO→OH, as a result, the RO$_2$

production would be weakened due to the critical production pathway of OH+VOCs→HO$_2$. In general, the

decreases in OH and RO$_2$ concentrations caused by HCHO were mainly dominated by the reduction of radical

propagation efficiencies. Meanwhile, in addition to the HCHO photolysis, radical propagation also played a

very important role in HO$_2$ production. In Zeng et al. (2019) study, HCHO had a non-negligible contribution

to HO$_2$ production, and the HO$_2$ production rates from HCHO photochemical reactions accounted for 15% of

total HO$_2$ production rates.



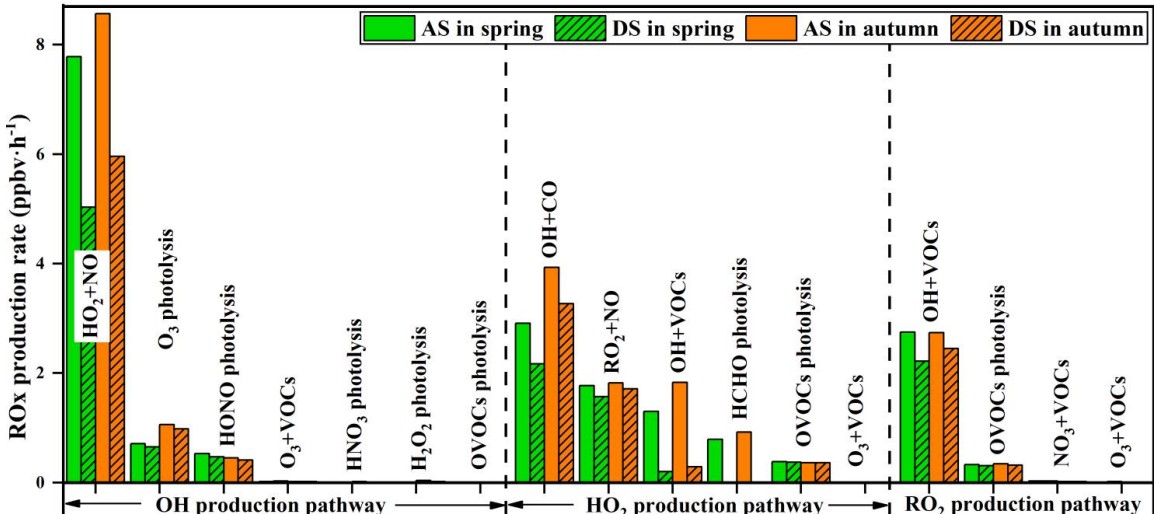

**Figure 10. The average daytime (06:00-17:00) production rates of OH, HO₂, and RO₂ in model scenarios**
**of AS and DS. AS scenario was run with all MCM mechanism, and DS scenario was run with the HCHO**
**mechanism disabled in MCM mechanism.**

**3.4.4 Impacts on the formation of O₃**
To investigate the impacts of HCHO on $O_3$ formation during the observation period, the detailed $O_3$
production and loss pathways in both AS and DS were quantified (Fig. 11 and Table 1). The daytime
production rates of $HO_2+NO$ and $RO_2+NO$ in AS were 7.78 and 2.96 ppbv h⁻¹ in spring and 8.57 and 2.87
ppbv h⁻¹ in autumn, accounting for 72% and 28% in spring and 75% and 25% in autumn of the total $O_3$
production, respectively. Meanwhile, $OH+NO_2$ was the predominant $O_3$ loss reaction with 1.93 ppbv h⁻¹ (60%)
in spring and 1.79 ppbv h⁻¹ (60%) in autumn, followed by $O_3$ photolysis (22% in spring and 29% in autumn),
$RO_2+NO_2$ (8% and 7%), $O_3+HO_2$ (3% and 8%), and $O_3+OH$ (3% and 6%), while the contributions of
$O_3+VOCs$ and $NO_3+VOCs$ pathways were very limited.

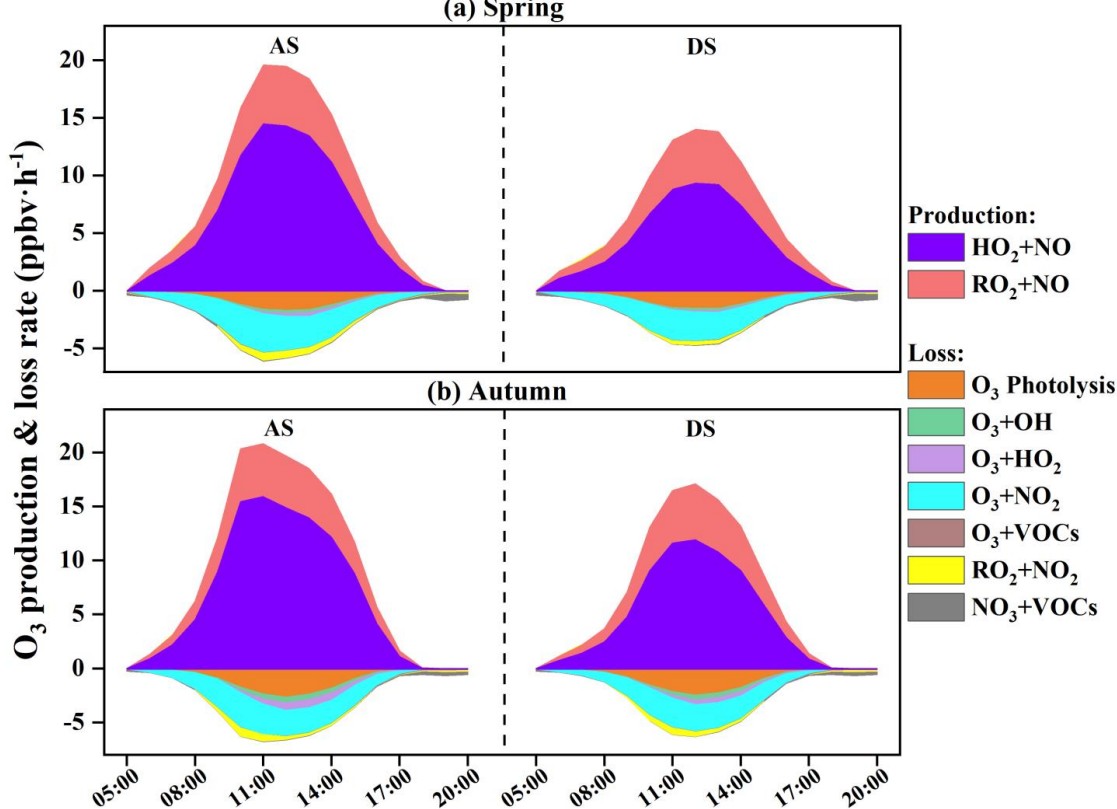


**Figure 11. Simulated profiles of O$_3$ mechanism in AS and DS in (a) spring and (b) autumn. AS scenario was run with all MCM mechanism, and DS scenario was run with the HCHO mechanism disabled in MCM mechanism.**

According to the differences between AS and DS, disabling HCHO mechanism could reduce the production rates of HO$_2$+NO by 2.75 ppbv h$^{-1}$ (35%) in spring and 2.61 ppbv h$^{-1}$ (30%) in autumn, and decrease RO$_2$+NO by 0.39 ppbv h$^{-1}$ (13%) in spring and 0.18 ppbv h$^{-1}$ (6%) in autumn (Table 1). About the O$_3$ loss pathways, the differences mainly reduced the rates of OH+NO$_2$ by 0.40 ppbv h$^{-1}$ (21%) in spring and 0.24 ppbv h$^{-1}$ (13%) in autumn, decreased O$_3$ photolysis by 0.06 (9%) in spring and 0.08 ppbv h$^{-1}$ (8%) in autumn, weakened RO$_2$+NO$_2$ by 0.16 ppbv h$^{-1}$ (62%) in spring and 0.05 ppbv h$^{-1}$ (19%) in autumn, lessened O$_3$+HO$_2$ by 0.06 ppbv h$^{-1}$ (50%) in spring and 0.12 ppbv h$^{-1}$ (43%) in autumn, and dropped O$_3$+OH by 0.03 ppbv h$^{-1}$ (31%) in spring and 0.04 ppbv h$^{-1}$ (19%) in autumn. The daytime average values of net O$_3$ production rates decreased by 32% in spring and 29% in autumn, and the peak net O$_3$ rate decreased by 32% in spring and 23% in autumn. Other studies indicated that the peak net O$_3$ rates in summer were reduced by 31% in a roadside site, 17% in a typical urban site, and 13% in a suburban site through diminishing HCHO (Zeng et al., 2019). When the HCHO mechanism was disabled, HO$_2$+NO, RO$_2$+NO$_2$, O$_3$+HO$_2$, O$_3$+OH, and OH+NO$_2$





showed significant changes. These reactions were all radical propagation pathways, which were consistent
with the results in Section 3.4.2. Therefore, the results highlighted and quantified the important impacts of
HCHO on $O_3$ formation in the southeast coastal area.

**Table 1. The average daytime (06:00-17:00) production rates, loss rates, and differences of $O_3$ in model**
**scenarios of AS and DS. AS scenario was run with all MCM mechanism, and DS scenario was run with**
**the HCHO mechanism disabled in MCM mechanism.**

| Reactions | Spring | | | Autumn | | |
|---|---|---|---|---|---|---|
| | AS | DS | Difference | AS | DS | Difference |
| $O_3$ production rate (ppbv h$^{-1}$) | | | | | | |
| $HO_2+NO$ | 7.78 | 5.03 | 35% | 8.57 | 5.96 | 30% |
| $RO_2+NO$ | 2.96 | 2.57 | 13% | 2.87 | 2.69 | 6% |
| $O_3$ loss rate (ppbv h$^{-1}$) | | | | | | |
| $OH+NO_2$ | 1.93 | 1.53 | 21% | 1.79 | 1.55 | 13% |
| $O_3$ photolysis | 0.71 | 0.65 | 9% | 1.06 | 0.98 | 8% |
| $RO_2+NO_2$ | 0.26 | 0.10 | 62% | 0.25 | 0.20 | 19% |
| $O_3+HO_2$ | 0.11 | 0.05 | 50% | 0.28 | 0.16 | 43% |
| $O_3+OH$ | 0.11 | 0.08 | 31% | 0.22 | 0.18 | 19% |
| Net $O_3$ production rate | 7.53 | 5.09 | 32% | 7.75 | 5.49 | 29% |


**4 Conclusions**
Combined field observations with model analyses were carried out in spring and autumn, when
photochemical pollution events frequently appeared in a coastal city of southeast China. We found that the
average levels of secondary products in autumn, such as $O_3$ and HCHO, were higher than those in spring,
relating to the intense photochemical reaction and meteorological conditions, although the concentrations of
NOx and VOCs in autumn were lower than those in spring. HCHO from secondary formation made the largest
contributions to ambient HCHO, followed by the primary sources of vehicle exhaust and biogenic emission.
The sensitivity analysis found that alkenes and aromatics were the most important precursors to HCHO from
secondary formation. Meanwhile, the top 5 precursors at the species level contributing to HCHO were propene,
ethene, toluene, m/p-xylene, and trans-2-butene, which were mainly emitted from combustion sources and
solvents use. The results indicated that the reduction of the two sources could effectively decrease both primary
and secondary sources of HCHO. Based on the analysis of disabling HCHO mechanism, we verified that
HCHO contributed to the AOC of 8% in spring and 10% in autumn, and decreased the concentrations of ROx,
reflecting the significance of HCHO in photochemistry. The daytime average values of net $O_3$ production rates
decreased by 32% in spring and 29% in autumn by disabling the HCHO mechanism. For the $O_3$ formation
mechanism, disabling HCHO mechanism reduced the production rates of $HO_2+NO$ and $RO_2+NO$, and
lessened the $O_3$ loss pathways of $OH+NO_2$, $RO_2+NO_2$, $O_3+HO_2$, and $O_3+OH$, indicating that the HCHO
affected $O_3$ formation mechanism mainly by controlling the efficiencies of radical propagation. This study





gives a scientific reference for HCHO source, formation pathway, and its contribution to the photochemistry
and further understanding of ozone pollution prevention in the coastal region.

**Code and Data availability**
The observation data at this site are available from the authors upon request.

**Authorship Contribution Statement**
Taotao Liu collected the data, contributed to the data analysis and performed chemical modeling analyses
of OBM-MCM, and wrote the paper. Jinsheng Chen designed the manuscript and supported funding of
observation and research. Yiling Lin collected the data and contributed to the PMF analysis. Youwei Hong
revised the manuscript. Gaojie Chen, Chen Yang, Lingling Xu, Mengren Li, Xiaolong Fan, Yanting Chen,
Liqian Yin, Yuping Chen, Xiaoting Ji, Ziyi Lin contributed to discussions of results. Fuwang Zhang and Hong
Wang provided part of the data in Xiamen.

**Competing interests**
The contact author has declared that neither they nor their co-authors have any competing interests.

**Acknowledgment**
This study was funded by the Cultivating Project of Strategic Priority Research Program of Chinese
Academy of Sciences (XDPB1903), the National Key Research and Development Program (2016YFC02005),
the National Natural Science Foundation of China (U1405235), the foreign cooperation project of Fujian
Province (2020I0038), the Xiamen Youth Innovation Fund Project (3502Z20206094), the FJIRSM&IUE Joint
Research Fund (RHZX-2019-006), center for Excellence in Regional Atmospheric Environment project
(E0L1B20201), Xiamen Atmospheric Environment Observation and Research Station of Fujian Province, and
Fujian Key Laboratory of Atmospheric Ozone Pollution Prevention (Institute of Urban Environment, Chinese
Academy of Sciences).





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
