# Peer review of "Seasonal characteristics of atmospheric formaldehyde (HCHO) in a coastal city of"

_Atmospheric Chemistry and Physics, 2022_

## Referee Comment (RC2)

Review of "Seasonal characteristics of atmospheric formaldehyde (HCHO) in a coastal city of southeast China: Formation mechanism and photochemical effects," Liu et al., ACP (2022)

**Summary**

This manuscript describes a set of ground-based observations of atmospheric composition at a coastal urban site in China. The primary analysis focus is formaldehyde (HCHO). Measurements are fed into a PMF model and a photochemical box model to estimate the sources of HCHO and the contributions of HCHO to radical chemistry and ozone production.

The reviewer has substantial concerns regarding the quality of HCHO observations, the interpretation of the PMF and box model results, and the general presentation of data and analysis. Many superfluous details are provided in the text. Text is highly descriptive without drawing out any obvious novel/new conclusions. This is a potentially useful contribution that hopefully will benefit from a hard critique. I recommend rejection with encouragement to resubmit.

**General Comments**

Regarding the HCHO analyzer described in Sect. 2.1: The reviewer was not able to locate any information about this analyzer on the internet, and there is no citation of literature regarding the design or performance of this instrument. The stated performance is 1 Hz, 50 pptv detection limit, 5% accuracy. This exceeds, by far, similar Hantzch-based instruments. For example, Glowania et al. (2021) report a 90-second time response, 300 pptv detection limit, and 8.6% accuracy (https://doi.org/10.5194/amt-14-4239-2021). Given that HCHO is central to this paper, additional documentation regarding calibration procedures and determination of potential artifacts is warranted.

Regarding interpretation of a highly-constrained model: Throughout the text, attention is given to the difference between HCHO production and loss rates (described as "net production rate" on L181). The model, however, is forced to measured HCHO. How well does the model predict HCHO if this constraint is turned off? If the model performs poorly, this calls into question the utility of the "net production rate" since the HCHO concentration does not match what would be predicted based on the modeled gross production rate.

PMF analysis: Several questions here.

1. Why are other species not included in PMF (CO, NOx, PAN)? In particular, CO should be a clear marker of vehicle exhaust.
2. Are the authors really suggesting that the HCHO associated with isoprene is directly emitted by the ecosystem? Is there any literature evidence of that? It seems more likely that this HCHO was produced by isoprene enroute to the site. Possibly without significant ozone production (e.g. from a nearby forest).
3. What is the real meaning of "secondary formation"? Again, it seems likely that the HCHO from those other sources is a mix of primary and secondary. It seems more accurate to call it "Ozone-associated HCHO." This is a general shortcoming of using PMF to parse something like HCHO and it should be acknowledged and clarified.

Data and Code Availability: According to FAIR standards, the observations and box model code should be publicly available without having to request them from the author.

**Specific Comments**

L24: The method for determining HCHO contributions should be stated here.

L59 – 62: suggest deletion of this sentence.

L146: why is the error factor 10% instead of actual measurement accuracy?

L166: How is the boundary layer height determined?

L173: updating constraints at hourly intervals is too coarse and likely leads to model artifacts due to step changes in photolysis and other parameters. 10 – 15 minute time steps are more appropriate for science-grade simulations.

Figure 2: There is little utility in showing atmospheric pressure and all 3 J's. You could remove the bottom panels and replace pressure in the top panels with shaded J(NO2).

L174: JNO2 is strongest in the visible, so applying a scaling factor from this variable alone may not capture variations in the UV (e.g. due to aerosol). How well does this JNO2 parameterization predict other measured J's, like JO1D or JHCHO?

L182: Why 20%? Are RIR values sensitive to this choice? Why not a smaller value (like 1%) so that radical resulting perturbations are locally linear?

L222: What other data supports "replenishment of HCHO primary emissions and accumulation of pollutants"?

L291: "under the intense solar radiation" is vague. Be quantitative. Based on Fig. 3, the J values are higher by 20% in autumn.

L29: The net production rate seems at odds with diurnal cycle of dHCHO/dt. At times, the observations show increasing HCHO when the model predicts loss, and vice versa. This is consistent with my second general comment about the model being over-constrained.

L301: CH3O2 comes from many precursors, so it is not quite fair to distinguish this from other RO2 precursors. This should be somehow stated or made clear, that the "RO + O2" bars in Fig. 5 are lower limits.

L312: "significantly higher" is not quantitative. Also, the loss rate might be higher because HCHO is higher. What is the difference in HCHO lifetimes between the two periods?

Sect. 3.4.1: This section does not describe the impacts of HCHO on atmospheric oxidation.

L421: Autumn OH is higher than any previous observations in China, at least among those cited. It is not fair to say the simulated HOx is "comparable." This is a lot of OH. Can any model comparisons be done to observations to substantiate it?

L430: Here, and elsewhere throughout the paper (L454, L501), are long lists of numbers that don't convey anything meaningful to the reader.

Figure 9: Figure S4 is potentially a more useful figure.

L449: If the model were truly constrained to ozone, there would be no decrease in ozone photolysis between these two runs. As alluded to in the General Comments, this is potentially an artifact of hourly time steps.

**Technical Comments**

English throughout would benefit from substantial copyediting.

Acronyms should only be defined after their first use.

Throughout the text, numbers are reported with too many significant figures. For example, 2.94 +/- 1.28 ppbv should be 2.9 +/- 1.3 ppbv. Also, it is often unclear what the averages and uncertainties/variabilities refer to (averaged of what period, at what time resolution).

---

## Author Comment (AC1)

**Response to Reviewers**

**Comment on acp-2022-292**

**Referee #1: Ye, Chunxiang**

**General comments:**

This manuscript reported relatively comprehensive measurements of HCHO, HONO, PAN, NOx, GC-FID/MS VOCs etc in city of Xiamen in Southeast China and presented source appointment analysis of HCHO by PMF model and simulation of oxidative capacity of the atmosphere by chemical model. First of all, I personally for now can not accept the PMF model source appointment of a relative short-lived species of multiple sources, like HCHO. This point surely needs further discussion. For example, the author can provide justification of the method, which is not available in current manuscript. Secondly, scientific motivation and measurement validation, or at least uncertainty analysis are two lost parts in discussion of the chemical model simulation. Otherwise, the result might appear to be of local interests or not robust, which is out of the scope of ACP journal.

**Response:** We appreciate Professor Ye for the positive comments and helpful suggestions. In the revised manuscript, we have addressed all of the comments, and particularly adopted the suggestion to synthesize our results and compare against existing findings of previous studies. The manuscript has been significantly revised and improved based on these suggestions. For clarity, the reviewer's comments are listed below in black, while our responses and changes in the manuscript are shown in blue and red respectively. We answer your questions about the (1) PMF model and (2) OBM-MCM model.

**(1)PMF model**

We agree with your idea that PMF model analysis of a relatively short-lived species requires caution, and we are afraid of the brief model introduction may cause confusion. Previous studies showed that photochemical processes could lead to the deviation between the primary and secondary sources of HCHO by the PMF model (Yuan et al., 2012b; Chen et al., 2014). Yuan et al. (2012) proved the capacity of the PMF approach in identifying the role of chemical aging for better understanding the PMF factors. The VOCs emission ratios derived from the PMF fresh factors agreed well with the ones calculated based on photochemical ages, indicating that the PMF approach could identify the contributions from primary emissions reasonably. Additionally, the abundances of NMHCs in the PMF-aged factors could be reproduced by the photochemical aging of fresh factors. In this study, we run PMF models and discussed the apportionment results to investigate whether the PMF approach can separate carbonyl sources well.

To further evaluate the performance of PMF simulation, the relationship between the contribution of one factor to each species and its chemical reactivity ($K_{OH}$) was analyzed to separate factors that are associated with fresh and aged emissions (Chen et al., 2014; Yuan et al., 2012). If the factor was related to fresh emissions, the distribution of each species would not correlate with its chemical reactivity, the HCHO distribution profile would be similar to that of each species in that factor. As the air mass from a source was aging, the NMHCs underwent photochemical reactions, and the more reactive species would be more largely consumed, while the HCHO distribution profile would be higher than each species in the aged factor due to the secondary production. As a result, a negative correlation could be seen between the aged factor contributions to each NMHC species and their $K_{OH}$ value. Figure 5 showed relationships between the factor contributions to each species and $K_{OH}$ values (representing chemical activities) for species. Indeed, the contributions of HCHO were higher than that of other species in factor 1 of secondary formation (the aged factor in Fig. 5a), but lower than or similar to other species in factor 2, 3, and 5 (the fresh factors in Fig. 5b, c, e). The factor 4 of biogenic source were thought as the aged factor because of the isoprene precursor for HCHO, thus the contributions of HCHO were relatively higher than other species (except isoprene) (Fig. 5d), furtherly confirming that PMF could reasonably identify the contributions of primary and secondary sources of HCHO (Chen et al., 2014). Anymore, the good correlations between the PMF model predicted and observed concentrations of each species were shown in Table S6 in Supplement. The correlation coefficients ($R^2$) were 0.76 for HCHO, 0.92 for $O_3$, 0.89 for 1,2-dichloroethane, and in the range of 0.60–0.99 for NMHC species. The PMF model is a multivariate factor analysis tool that also has been repeatedly applied to the source identification of carbonyls (Zeng et al., 2019; Chen et al., 2014; Guo et al., 2013; Ling et al., 2017; Yuan et al., 2012).

The relevant contents of the justification of the PMF method were added to our manuscript, as follows:

"Yuan et al. (2012) proved the capacity of the PMF approach in identifying the role of chemical aging, and the abundances of NMHCs in the PMF-aged factors could be reproduced by the photochemical aging of fresh factors. Considering the photochemical process impacts of the PMF factors, the relationship between the contribution of one factor to each species and its chemical reactivity ($K_{OH}$) was analyzed to separate factors that are associated with fresh and aged emissions (Chen et al., 2014; Yuan et al., 2012). The distribution of each species in the fresh factor would not correlate with its chemical reactivity, thus the HCHO distribution profile would be similar to that of each species in that factor. In the aged factors, the NMHCs with more chemical activities would be more largely consumed by photochemical reactions, while the HCHO distribution profile would be higher than each species due to the secondary production. Figure 5 showed relationships between the factor contributions to each species and $K_{OH}$ values (representing chemical activities) for species. Indeed, the contributions of HCHO were higher than that of other species in factor 1 of secondary formation (the aged factor in Fig. 5a), but lower than or similar to other species in factor 2, 3, and 5 (the fresh factors

in Fig. 5b,c, e). The factor 4 of biogenic source were thought as the aged factor because of the isoprene precursor for HCHO, thus the contributions of HCHO were relatively higher than other species (except isoprene) (Fig. 5d), further confirming that PMF reasonably identified the contributions of primary and secondary sources of HCHO (Chen et al., 2014). In this study, we run PMF models and discussed the apportionment results indicating the PMF approach can separate HCHO sources well."

[Figure]

**Figure 5. Relationship between the factor contributions to each species and $K_{OH}$ values(representing chemical activities) of the species. Each square represents one species, while HCHO is represented as a square in red.**

**Table S6. Correlation coefficients ($R^2$) between observed and simulated concentrations of each species**

| Species | $R^2$ |
|---|---|
| $O_3$ | 0.92 |
| ethene | 0.90 |
| acetylene | 0.84 |
| ethane | 0.93 |
| propene | 0.86 |
| propane | 0.77 |
| iso-butane | 0.84 |
| 1-butene | 0.87 |
| n-butane | 0.88 |
| iso-pentane | 0.78 |
| n-pentane | 0.93 |
| isoprene | 0.99 |
| 3-methylpentane | 0.81 |
| benzene | 0.85 |
| toluene | 0.86 |
| m/p-xylene | 0.60 |
| ethylbenzene | 0.69 |
| o-xylene | 0.63 |
| formaldehyde | 0.76 |
| 1,2-dichloroethane | 0.89 |

**(2) the chemical model of OBM-MCM**

We agree with your idea that measurement validation and model performance evaluation are essential. Figure S1 shows the simulated and observed HCHO at the study site. According to previous studies, the inconsistency between simulated and observed HCHO could be caused by the uncertainties in the treatment of dry deposition, faster vertical transport, uptake of HCHO, atmospheric diffusion/dilution meteorological conditions, and fresh emission of precursor VOCs (Li et al., 2014; Zhang et al., 2021). The differences between the modeled HCHO concentrations and observed concentrations frequently is used to judge the rationality of the model results, and the methods were named the index of agreement (IOA), which were calculated by the equation (Zhang et al., 2021):

$$IOA = 1 - \frac{\sum_{i=1}^{n}(O_i - S_i)^2}{\sum_{i=1}^{n}(|O_i - \bar{O}| - |S_i - \bar{O}|)^2} \qquad (1)$$

Where, $S_i$ is modeled HCHO value, $O_i$ represents observed HCHO concentration, $\bar{O}$ is the average observed HCHO value, and n is the sample number. The IOA range is 0-1, and the higher the IOA value is, the better agreement between modeled and observed values is. In many studies, IOA ranges from 0.68 to 0.89 (Wang et al., 2018), and the

modeled results are reasonable. The IOAs in spring and autumn in our research are 0.83 and 0.80, respectively. Hence, the modeled and observed HCHO during the observation periods at the study site matched the true situation, and the performance of the OBM-MCM model was reasonably acceptable. Although there is a certain discrepancy, the model could generally reflect the atmospheric chemical processes, and these results still provide valuable information on secondary formation of HCHO at our study site. And this part of the model validation of IOA also has been added to the supplement materials of Text 1.

[Figure]

**Figure S1. The simulated and observed HCHO at the study site.**

**Specific comments**

I I found some puzzled statement. For example,

In line 19ï¼š The author emphasis experiment design in a coastal area. However, what is special for this area? Any implication for general coastal area from this dataset?

**Response:** Xiamen is located in the coastal region of Southeast China under the East Asian monsoon control, belonging to the subtropical marine climate (Liu et al., 2020a, b). In spring, north cold airflow and south warm airflow formed the quasi-stationary front causing atmospheric stagnation. In autumn, under the control of the West Pacific subtropical high (WPSH), favorable meteorological conditions enhanced the formation and accumulation of photochemical pollutants (Wu et al., 2020). Meanwhile, the southeastern coastal region is influenced by the East Asian monsoon and acts as an important transport path between the Yangtze River Delta (YRD) and the Pearl River Delta (PRD) (T. Liu et al., 2020a, b). In Xiamen, the concentrations of $O_3$ precursors were higher than those in remote sites and backgrounds but lower than those in most urban and suburban areas, even lower than those in rural regions (Table R1). Anymore,

our previous study showed that atmospheric particles were low in Xiamen (Hu et al., 2022; Deng et al., 2020). Despite the few pollution sources, Xiamen frequently appeared $O_3$ pollution events in spring and autumn, when the meteorological conditions were governed by weather systems such as the quasi-stationary front and the west pacific subtropical high (Liu et al., 2022a; Wu et al., 2019). The favorable photochemical reaction conditions (including high air temperature, low relative humidity, intense solar radiation, and stagnant atmosphere) provided a good 'laboratory' to further explore the HCHO formation mechanism and its impact on $O_3$ formation.

**Table R1. Comparison of NO, $NO_2$, and total VOCs levels in cities between China and other countries (Units: ppbv).**

| Location | $NO_2$ | NO | VOCs | Site category | Observation periods | Reference |
|---|---|---|---|---|---|---|
| Xiamen | 9.8 | 3.7 | 23.9 | Urban | May.-Jun. 2021 | This study |
| Xiamen | 8.6 | 2.2 | 17.4 | Urban | Sep. 2021 | This study |
| Beijing | 16.8 | 2.1 | 44.2 | Urban | | |
| Wuhan | 17.5 | 3.2 | 30.2 | Urban | Summer 2018 (episode) | Liu et al., 2021b |
| Lanzhou | 15.8 | 2.9 | 45.3 | Urban | | |
| Shanghai | 14.2 | 3.38 | 25.3 | Urban | Jun. 2019 (episode) | Zhu et al., 2020 |
| Chengdu | 39.0 | 3.6 | 36.0 | Urban | Jul. 2017 (episode) | Yang et al., 2020 |
| Los Angeles | - | - | 41.3 | Urban | May.–Jun. 2010 | Warneke et al., 2012 |
| London | - | - | 22.1 | Urban | 1998–2008 | Von Sch. et al., 2010 |
| Tokyo | - | - | 43.4 | Urban | 2003–2005 | Hoshi et al., 2008 |
| Beijing | 11.5 | 4.8 | 28.1 | Suburban | Aug. 2018 | Yang et al., 2021 |
| Hong Kong | 25.0 | 14.0 | 26.9 | Suburban | Aug. to Nov. 2013 | Wang et al., 2018 |
| Chengdu | 11.4 | 8.0 | 28.0 | Suburban | Summer 2019 | Yang et al., 2021a |
| Qingdao | 16.7 | 1.6 | 7.6 | Rural | Oct.–Nov. 2019 | Liu et al., 2021a |
| The Pearl River Delta | 39.9 | 4.2 | 38.0 | Rural | Octo.–Nov. 2014 | He et al., 2019 |
| Hong Kong | 12.2 | 1.9 | 10.9 | Regional background | Aug.–Dec. 2012 | Li et al., 2018 |
| Mt. Wuyi | - | - | 4.7 | Background | Dec. 2016 | Hong et al., 2019 |
| Mt. Tai | - | - | 8.8 | Background | Jun. 2006 | Suthawaree et al., 2010 |
| Mt. Waliguan | - | - | 2.6 | Remote region | Jul.–Aug. 2003 | Xue et al., 2013 |

Note: "-" means that the data was not mentioned in the relevant studies.

The AOC levels depend on solar radiation, concentrations, and types of air pollutants in different sites. Table R2 shows the precursor concentration, AOC, and $JNO_2$ levels in Xiamen, Shanghai, and Hong Kong. Due to the characteristics of our study site, although the levels of $O_3$ precursors in these urban sites were higher than those in Xiamen, the photolysis rates in these cities were lower than those in Xiamen. Hence, the recalculated AOC level in this study was comparable to those at urban sites in Hongkong and Shanghai we mentioned.

**Table R2. Comparison of NO, NO₂, total VOCs (ppbv), AOC (molecules cm⁻³ s⁻¹), and J(NO₂) levels in Xiamen, Shanghai, and Hong Kong.**

| Location | $NO_2$ | NO | VOCs | Site category | AOC | Maximum AOC | $JNO_2$ $(10^{-3} s^{-1})$ | Maximum $JNO_2$ | Reference |
|---|---|---|---|---|---|---|---|---|---|
| Xiamen | 8.6 | 2.2 | 17.4 | Urban | $6.7 \times 10^7$ | $1.5 \times 10^8$ | 3.4 | 11.5 | This study |
| Shanghai | 14.2 | 3.4 | 25.3 | Urban | $3.9 \times 10^7$ | $1.0 \times 10^8$ | 2.8 | 8.0 | Liu et al., 2020 |
| Hong Kong | - | - | 32.7 | Urban | $6.3 \times 10^7$ | $1.3 \times 10^8$ | - | 6.0 | Xue et al., 2016 |
| Hong Kong | 12.2 | 1.9 | 10.9 | Regional background | $1.6 \times 10^7$ | $6.2 \times 10^7$ | 2.3 | 9.3 | Li et al., 2018 |

Note: "-" means that the data was not mentioned in the relevant studies.

Moreover, the favorable photochemical reaction conditions (including high air temperature, low relative humidity, intense solar radiation, and stagnant atmosphere) in our study site promoted the photochemical reaction of HCHO. Hence, HCHO photochemical rates were higher than those in some urban/suburban sites with similar precursor levels, and comparable to those in some polluted regions, but the daytime average HCHO loss rate was lower than that in these sites (Table R3 and Figure R1). the intensive solar radiation in our study accelerated HCHO loss rate, but relatively low precursor levels limited HCHO secondary formation (Zhang et al., 2021a; Yang et al., 2020). The net HCHO production rate during the daytime (06:00-17:00 LT) in autumn (−0.4±0.7 ppbv h⁻¹) was lower than that in spring (0.1±0.4 ppbv h⁻¹), and a negative net HCHO production indicates that strong photochemical reactions can constrain high HCHO levels in certain situations.

The findings of this study provide significant guidance for emission reduction and regional collaboration for future photochemical pollution control in the relatively clean coastal cities of China and similar countries. We have also revised relevant contents in the manuscript to highlight the characteristics of our findings.

**Table R3. Comparison of HCHO (ppbv) and HCHO production and loss rates (ppbv·s⁻¹) in Xiamen, Wuhan, Shanghai, and Hong Kong.**

| Location | HCHO | Site category | HCHO production rate | HCHO loss rate | Reference |
|---|---|---|---|---|---|
| Xiamen | 2.9 | Urban-spring | 1.92 | 1.88 | This study (Fig. R1 a) |
| | 3.2 | Urban-autumn | 2.01 | 2.48 | |
| Shanghai | 6.7 | Suburban | 3.3 | 2.6 | Zhang et al., 2021 (Fig. R1 b) |
| | 2.2 | Suburban (WJS) | 1.2 | 1.02 | |
| Wuhan | 2.1 | Urban (ZY) | 0.8 | 0.6 | Zeng et al., 2019 (Fig. R1 c) |
| | 3.4 | Roadside (HK) | 0.33 | 0.15 | |
| Hong Kong | 2.0 | Regional background | 0.75 | 0.63 | Yang et al., 2020 (Fig. R1 d) |

[Figure]

**Figure R1. Average production and destruction pathways of HCHO at (a) Xiamen, (b) Shanghai, (c)Wuhan of sites in WJS, ZY, and HK, and (d) Hong Kong. Wujiashan (WJS) (30.64° N, 114.21° E) is a suburban site settled on the top floor (~12m above ground level (a.g.l)) of a high school and surrounded by residential apartments. ZY (30.55° N, 114.30° E) is a typical urban site located in a mixed commercial and residential area in the center of Wuhan. HK (30.60° N, 114.27° E) is a roadside site.**

In line 18-19: whether measurements of HCHO is scarce or not?

**Response:** We are sorry for the confusion caused by the unclear description. Numerous studies have reported on the serious HCHO pollution in China, most of which mainly focused on either its pollution levels or potential sources in the polluted regions of China (Li et al., 2010; Ling et al., 2017; Yang et al., 2017; Yang et al., 2019; Cheng et al., 2014; Guo, 2009; Louie et al., 2013). In recent, limited studies have focused on the formation mechanisms of HCHO in the megacities and regions with rapid economic development or with severe pollution in China, such as Beijing (Liu et al., 2015; Yang et al., 2018) and Pearl River Delta (PRD) (Li et al., 2014; Ling et al., 2017). Yet, to the best of our knowledge, the HCHO budget in the Xiamen region remains unclear currently. In recent years, few studies quantified the HCHO photochemical reactions and the impacts of HCHO on atmospheric photochemistry (Zhang et al, 2021a; Zeng et al, 2019), and the relevant researches are still scarce in southeast coastal areas of China. Different types and sources for HCHO precursors, as well as the influences of weather and meteorological conditions, lead to the complicated secondary formation mechanisms in various regions, thus the exploration of HCHO sources and photochemical effects are very necessary for ozone pollution mitigation by efficient control strategies. To avoid misunderstanding, we have revised the relevant expressions in the manuscript, as follows:

"Yet, the current studies on quantification of HCHO impacts on atmospheric photochemistry in southeast coastal areas remain scarce and unclear,"

I would say the role of HCHO in radical chemistry is relative well accepted. If the author suggest otherwise, what is new in the understanding of this subject from your data?

**Response:** We thank the reviewer very much for the critical comments which would definitely help us to improve our work.

Firstly, our study is based on a full suite of field observations, then an observation-based photochemical model was adopted as the main research tool. Our observations of multi-parameters were based on the Atmospheric Environment Observation Supersite (AEOS, 24.61° N, 118.06° E), which is an atmospheric environment observation platform with multi-parameter and high temporal resolution, especially the analyzers of HCHO (FMS-100, Focused Photonics Inc., Hangzhou, China), PAN (PANs-1000, Focused Photonics Inc., Hangzhou, China) and HONO (MARGA, ADI 2080, Applikon Analytical B.V., the Netherlands), to find out the distribution law and the characteristics of the important active species that affect the photochemical pollution mechanism of the atmosphere in the study area significantly. Meanwhile, the photolysis frequencies are the driving force and main controlling factor of atmospheric photochemical process. In our study, eight key photolysis frequencies (i.e. $J$HCHO, $J$O$^1$D, $J$NO$_2$, $J$HONO, $J$H$_2$O$_2$, and $J$NO$_3$) are monitored by a photolysis spectrometer (PFS-100, Focused Photonics Inc., Hangzhou, China) in real-time. Because of lacking systematic field measurement, many previous researchers only used basic data (VOCs, criteria air pollutants of O$_3$, NOx, and CO) to run the model, and the missing active species did not constrain the model or were substituted with parameters (such as HONO was fixed to 0.02 of the observed NO$_2$ levels), while the photolysis frequencies would be estimated using the Tropospheric Ultraviolet and Visible Radiation (TUV) model (http://cprm.acom.ucar.edu/Models/TUV/) (Tan et al., 2019; Jiang et al., 2020). The lack of monitoring values of these important parameters will greatly increase the uncertainty of the model. Hence, the relatively systematic observed data can well optimize and localize the model, which is the advantage of field observation.

Secondly, we just would like to state the rationale of this study, which is to quantify the reaction pathways of pollution mechanism and the contribution of HCHO to photochemistry based on the 'Known Chemistry' as well as comprehensive measurements of related species/parameters. We think these results should be helpful for better understanding the atmospheric oxidation chemistry in the relatively clean coastal city of Xiamen. As the answer to your first specific question shown, the southeastern coastal area has a long duration of sunshine throughout the year, intense solar radiation, and fast photochemical conversion rates. The local formation, regional transport, and offshore complex meteorological conditions were necessary conditions for photochemical pollution in our study site with relatively few pollution sources (Liu et al., 2022). In this study, the regional characteristics brought the differences in

photochemistry compared with other regions (such as the AOC and HCHO mechanism of the answer to your second specific question). Overall, these results should be useful for the community to understand the atmospheric chemistry in different metropolitan areas of the world. Xiamen, a typical city with an upward trend in ozone pollution, was selected as the study area to research the distribution of photochemical active species of HCHO and its impact on atmospheric chemistry, which is the advantage of regional characteristics.

.

Reference:

Chen, T., Xue, L., Zheng, P., Zhang, Y., Liu, Y., Sun, J., Han, G., Li, H., Zhang, X., Li, Y., Li, H., Dong, C., Xu, F., Zhang, Q., and Wang, W.: Volatile organic compounds and ozone air pollution in an oil production region in northern China, Atmospheric Chemistry and Physics, 20, 7069-7086, 10.5194/acp-20-7069-2020, 2020.

Chen, W. T., Shao, M., Lu, S. H., Wang, M., Zeng, L. M., Yuan, B., and Liu, Y.: Understanding primary and secondary sources of ambient carbonyl compounds in Beijing using the PMF model, Atmospheric Chemistry and Physics, 14, 3047-3062, 10.5194/acp-14-3047-2014, 2014.

Cheng, Y., Lee, S.C., Huang, Y., Ho, K.F., Ho, S.S.H., Yau, P.S., Louie, P.K.K., Zhang, R.J., 2014. Diurnal and seasonal trends of carbonyl compounds in roadside, urban, and suburban environment of Hong Kong. Atmos. Environ. 89, 43–51.

Deng, J., Guo, H., Zhang, H., Zhu, J., Wang, X., and Fu, P.: Source apportionment of black carbon aerosols from light absorption observation and source-oriented modeling: an implication in a coastal city in China, Atmos. Chem. Phys., 20, 14419–14435, https://doi.org/10.5194/acp-20-14419-2020, 2020.

Guo, H., 2009. Concurrent observations of air pollutants at two sites in the Pearl River Delta and the implication of regional transport. Atmos. Chem. Phys. 9, 7343–7360.

Guo, H., Ling, Z. H., Cheung, K., Wang, D. W., Simpson, I. J., and Blake, D. R.: Acetone in the atmosphere of Hong Kong: Abundance, sources and photochemical precursors, Atmospheric Environment, 65, 80-88, 10.1016/j.atmosenv.2012.10.027, 2013.

Hu, B., Duan, J., Hong, Y., Xu, L., Li, M., Bian, Y., Qin, M., Fang, W., Xie, P., and Chen, J.: Exploration of the atmospheric chemistry of nitrous acid in a coastal city of southeastern China: results from measurements across four seasons, Atmos. Chem. Phys., 22, 371–393, https://doi.org/10.5194/acp-22-371-2022, 2022.

Jiang, Y., Xue, L., Gu, R., Jia, M., Zhang, Y., Wen, L., Zheng, P., Chen, T., Li, H., Shan, Y., Zhao, Y., Guo, Z., Bi, Y., Liu, H., Ding, A., Zhang, Q., and Wang, W.: Sources of

nitrous acid (HONO) in the upper boundary layer and lower free troposphere of the North China Plain: insights from the Mount Tai Observatory, Atmospheric Chemistry and Physics, 20, 12115-12131, 10.5194/acp-20-12115-2020, 2020.

Li, X., Rohrer, F., Brauers, T., Hofzumahaus, A., Lu, K., Shao, M., Zhang, Y. H., and Wahner, A.: Modeling of HCHO and CHOCHO at a semi-rural site in southern China during the PRIDE-PRD2006 campaign, Atmospheric Chemistry and Physics, 14, 12291-12305, 10.5194/acp-14-12291-2014, 2014.

Li, Y., Shao, M., Lu, S., Chang, C.-C., Dasgupta, P.K., 2010. Variations and sources of ambient formaldehyde for the 2008 Beijing Olympic games. Atmos. Environ. 44, 2632–2639.

Ling, Z. H., Zhao, J., Fan, S. J., and Wang, X. M.: Sources of formaldehyde and their contributions to photochemical O3 formation at an urban site in the Pearl River Delta, southern China, Chemosphere, 168, 1293-1301, 10.1016/j.chemosphere.2016.11.140, 2017.

Ling, Z.H., Guo, H., Simpson, I.J., Saunders, S.M., Lam, S.H.M., Lyu, X.P., et al., 2016. New insight into the spatiotemporal variability and source apportionments of C1–C4 alkyl nitrates in Hong Kong. Atmos. Chem. Phys. 16, 8141–8156.

Liu, T., Hong, Y., Li, M., Xu, L., Chen, J., Bian, Y., Yang, C., Dan, Y., Zhang, Y., Xue, L., Zhao, M., Huang, Z., and Wang, H.: Atmospheric oxidation capacity and ozone pollution mechanism in a coastal city of southeastern China: analysis of a typical photochemical episode by an observation-based model, Atmospheric Chemistry and Physics, 22, 2173-2190, 10.5194/acp-22-2173-2022, 2022.

Liu, Y., Shen, H., Mu, J., Li, H., Chen, T., Yang, J., Jiang, Y., Zhu, Y., Meng, H., Dong, C., Wang, W., and Xue, L.: Formation of peroxyacetyl nitrate (PAN) and its impact on ozone production in the coastal atmosphere of Qingdao, North China, Sci Total Environ, 778, 146265, 10.1016/j.scitotenv.2021.146265, 2021.

Liu, Y., Yuan, B., Li, X., Shao, M., Lu, S., Li, Y., Chang, C.C., Wang, Z., Hu, W., Huang, X., He, L., Zeng, L., Hu, M., Zhu, T., 2015. Impact of pollution controls in Beijing on atmospheric oxygenated volatile organic compounds (OVOCs) during the 2008 Olympic Games: observation and modeling implications. Atmos. Chem. Phys. 15, 3045–3062.

Louie, P.K.K., Ho, J.W.K., Tsang, R.C.W., Blake, D.R., Lau, A.K.H., Yu, J.Z., Yuan, Z., Wang, X., Shao, M., Zhong, L., 2013. VOCs and OVOCs distribution and control policy implications in Pearl River Delta region. China. Atmos. Environ. 76, 125–135.

Lu K, Guo S, Tan Z, Wang H, Shang D, Liu Y, Li X, Wu Z, Hu M, Zhang Y: Exploring atmospheric free-radical chemistry in China: the self-cleansing capacity and the formation of secondary air pollution. National Science Review, 6(3):579-594, 2019.

Lu KD and Zhang YH. Observations of HO(x) Radical in field studies and the analysis

of its chemical mechanism. Prog Chem 2010; 22: 500–14.

Saunders, S. M., Jenkin, M. E., Derwent, R. G., and Pilling, M.J.: Protocol for the development of the Master Chemical Mechanism, MCM v3 (Part A): tropospheric degradation of nonaromatic volatile organic compounds, Atmos. Chem. Phys., 3, 161–180, https://doi.org/10.5194/acp-3-161-2003, 2003.

Tan, Z., Lu, K., Jiang, M., Su, R., Wang, H., Lou, S., Fu, Q., Zhai, C., Tan, Q., Yue, D., Chen, D., Wang, Z., Xie, S., Zeng, L., and Zhang, Y.: Daytime atmospheric oxidation capacity in four Chinese megacities during the photochemically polluted season: a case study based on box model simulation, Atmospheric Chemistry and Physics, 19, 3493-3513, 10.5194/acp-19-3493-2019, 2019.

Uria-Tellaetxe, I., and Carslaw, D. C.: Conditional bivariate probability function for source identification, Environ. Model. Softw., 59, 1-9, 2014.

Wang, H., Lyu, X., Guo, H., Wang, Y., Zou, S., Ling, Z., Wang, X., Jiang, F., Zeren, Y., Pan, W., Huang, X., and Shen, J.: Ozone pollution around a coastal region of South China Sea: interaction between marine and continental air, Atmos. Chem. Phys., 18, 4277–4295, 10.5194/acp-18-4277-2018, 2018.

Wang, M., Chen, W., Zhang, L., Qin, W., Zhang, Y., Zhang, X., and Xie, X.: Ozone pollution characteristics and sensitivity analysis using an observation-based model in Nanjing, Yangtze River Delta Region of China, J Environ Sci (China), 93, 13-22, 10.1016/j.jes.2020.02.027, 2020.

Wang, Q., Chen, W., Xu, W.T., Yang, P., Liu, Y.Y., Ma, J.Y., Hua, D.Z.: Formaldehyde Reaction Cell, CN212207248U, https://kns.cnki.net/kcms/detail/detail.aspx?FileName

=CN212207248U&DbName=SCPD2020, 2020.

Wen, L., Chen, T., Zheng, P., Wu, L., Wang, X., Mellouki, A., Xue, L., and Wang, W.: Nitrous acid in marine boundary layer over eastern Bohai Sea, China: Characteristics, sources, and implications, Sci. Total Environ., 670, 282–291, https://doi.org/10.1016/j.scitotenv.2019.03.225, 2019.

Yang, X., Xue, L., Wang, T., Wang, X., Gao, J., Lee, S., Blake, D.R., Chai, F., Wang, W., 2018. Observations and explicit modeling of summertime carbonyl formation in Beijing: identification of key precursor species and their impact on atmospheric oxidation chemistry: observations and explicit modeling of summertime carbonyl formation in Beijing: Ide. J. Geophys. Res. Atmos. 123.

Yang, X., Xue, L., Yao, L., Li, Q., Wen, L., Zhu, Y., Chen, T., Wang, X., Yang, L., Wang, T., Lee, S., Chen, J., Wang, W., 2017. Carbonyl compounds at Mount Tai in the North China Plain: characteristics, sources, and effects on ozone formation. Atmos. Res. 196, 53–61.

Yang, X., Zhang, G., Sun, Y., Zhu, L., Wei, X., Li, Z., and Zhong, X.: Explicit modeling of background HCHO formation in southern China, Atmospheric Research, 240, 104941, 10.1016/j.atmosres.2020.104941, 2020.

Yang, Z., Cheng, H.R., Wang, Z.W., Peng, J., Zhu, J.X., Lyu, X.P., Guo, H., 2019. Chemical characteristics of atmospheric carbonyl compounds and source identification of formaldehyde in Wuhan. Central China. Atmos Res. 228, 95–106.

Yuan, B., Shao, M., de Gouw, J., Parrish, D. D., Lu, S., Wang, M., Zeng, L., Zhang, Q., Song, Y., Zhang, J., and Hu, M.: Volatile organic compounds (VOCs) in urban air: How chemistry affects the interpretation of positive matrix factorization (PMF) analysis, Journal of Geophysical Research: Atmospheres, 117, n/a-n/a, 10.1029/2012jd018236, 2012.

Zeng, P., Lyu, X., Guo, H., Cheng, H., Wang, Z., Liu, X., and Zhang, W.: Spatial variation of sources and photochemistry of formaldehyde in Wuhan, Central China, Atmospheric Environment, 214, 116826, 10.1016/j.atmosenv.2019.116826, 2019.

Zhang, G., Hu, R., Xie, P., Lu, K., Lou, S., Liu, X., Li, X., Wang, F., Wang, Y., Yang, X., Cai, H., Wang, Y., and Liu, W.: Intercomparison of OH radical measurement in a complex atmosphere in Chengdu, China, Sci Total Environ, 838, 155924, 10.1016/j.scitotenv.2022.155924, 2022.

Zhang, K., Duan, Y., Huo, J., Huang, L., Wang, Y., Fu, Q., Wang, Y., and Li, L.: Formation mechanism of HCHO pollution in the suburban Yangtze River Delta region, China: A box model study and policy implementations, Atmospheric Environment, 267, 118755, 10.1016/j.atmosenv.2021.118755, 2021a.

Zhang, K., Huang, L., Li, Q., Huo, J., Duan, Y., Wang, Y., Yaluk, E., Wang, Y., Fu, Q., and Li, L.: Explicit modeling of isoprene chemical processing in polluted air masses in suburban areas of the Yangtze River Delta region: radical cycling and formation of ozone and formaldehyde, Atmospheric Chemistry and Physics, 21, 5905-5917, 10.5194/acp-21-5905-2021, 2021b.

---

## Author Comment (AC2)

**Response to Reviewers**

**Comment on acp-2022-292**

**RC2** Anonymous Referee #2**

Review of "Seasonal characteristics of atmospheric formaldehyde (HCHO) in a coastal city of southeast China: Formation mechanism and photochemical effects," Liu et al., ACP (2022)

**Summary**

This manuscript describes a set of ground-based observations of atmospheric composition at a coastal urban site in China. The primary analysis focus is formaldehyde (HCHO). Measurements are fed into a PMF model and a photochemical box model to estimate the sources of HCHO and the contributions of HCHO to radical chemistry and ozone production.

The reviewer has substantial concerns regarding the quality of HCHO observations, the interpretation of the PMF and box model results, and the general presentation of data and analysis. Many superfluous details are provided in the text. Text is highly descriptive without drawing out any obvious novel/new conclusions. This is a potentially useful contribution that hopefully will benefit from a hard critique. I recommend rejection with encouragement to resubmit.

**Response:** Thanks for your feedback on the whole manuscript and valuable comments on some details. The detailed introduction of HCHO observations was added to our manuscript, and the related issues of PMF and box model have also been explained and resolved. We have tried our best to improve the quality of this manuscript. Many analyses have been improved to be complete and easy to understand accordingly.

**General Comments**

Regarding the HCHO analyzer described in Sect. 2.1: The reviewer was not able to locate any information about this analyzer on the internet, and there is no citation of literature regarding the design or performance of this instrument. The stated performance is 1 Hz, 50 pptv detection limit, 5% accuracy. This exceeds, by far, similar Hantzch-based instruments. For example, Glowania et al. (2021) report a 90-second time response. 300 pptv detection limit, and 8.6% accuracy (https://doi.org/10.5194/amt-14-4239-2021). Given that HCHO is central to this paper, additional documentation regarding calibration procedures and determination of potential artifacts is warranted.

**Response:** Thanks for your suggestion. The reaction chamber system of our HCHO analyzer has optimizations compared to theirs (Glowania et al. 2021). The HCHO analyzer was customized by Hangzhou Focused Photonics Inc., of which the patent was announced in December 2020 (Wang et al., 2020) and there are currently no published researches to cite. Formaldehyde standard solution can react with Hantzsch reagent at room temperature with a slow reaction speed. In this study, stainless steel tube for heating efficiency and ceramic fiber board thermal insulation layer for thermal insulation efficiency were used to control the reaction temperature of the mixed solution in the reaction chamber to shorten the reaction time, which reduced the time required for thermal equilibrium significantly, achieving a signal acquisition frequency of 1 per second.

After completing the multi-point calibration, zero gas tests for more than 1 hour were carried out, then the standard deviations of 60 sets of gas concentration data were obtained, and the detection limit was 3 times the standard deviation. Table S1 shows the detection limit results in our study.

| Time                | HCHO (ppbv) | Time                | HCHO (ppbv) |
|---------------------|-------------|---------------------|-------------|
| 2021-05-14 01:06:58 | -0.168      | 2021-05-14 01:07:28 | -0.157      |
| 2021-05-14 01:06:59 | -0.167      | 2021-05-14 01:07:29 | -0.157      |
| 2021-05-14 01:07:00 | -0.166      | 2021-05-14 01:07:30 | -0.157      |
| 2021-05-14 01:07:01 | -0.165      | 2021-05-14 01:07:31 | -0.156      |
| 2021-05-14 01:07:02 | -0.164      | 2021-05-14 01:07:32 | -0.156      |
| 2021-05-14 01:07:03 | -0.163      | 2021-05-14 01:07:33 | -0.156      |
| 2021-05-14 01:07:04 | -0.162      | 2021-05-14 01:07:34 | -0.155      |
| 2021-05-14 01:07:05 | -0.161      | 2021-05-14 01:07:35 | -0.155      |
| 2021-05-14 01:07:06 | -0.161      | 2021-05-14 01:07:36 | -0.154      |
| 2021-05-14 01:07:07 | -0.161      | 2021-05-14 01:07:37 | -0.154      |
| 2021-05-14 01:07:08 | -0.161      | 2021-05-14 01:07:38 | -0.154      |
| 2021-05-14 01:07:09 | -0.16       | 2021-05-14 01:07:39 | -0.153      |
| 2021-05-14 01:07:10 | -0.16       | 2021-05-14 01:07:40 | -0.153      |
| 2021-05-14 01:07:11 | -0.159      | 2021-05-14 01:07:41 | -0.153      |
| 2021-05-14 01:07:12 | -0.159      | 2021-05-14 01:07:42 | -0.153      |
| 2021-05-14 01:07:13 | -0.158      | 2021-05-14 01:07:43 | -0.152      |
| 2021-05-14 01:07:14 | -0.158      | 2021-05-14 01:07:44 | -0.152      |
| 2021-05-14 01:07:15 | -0.158      | 2021-05-14 01:07:45 | -0.151      |
| 2021-05-14 01:07:16 | -0.158      | 2021-05-14 01:07:46 | -0.151      |
| 2021-05-14 01:07:17 | -0.158      | 2021-05-14 01:07:47 | -0.15       |
| 2021-05-14 01:07:18 | -0.158      | 2021-05-14 01:07:48 | -0.15       |
| 2021-05-14 01:07:19 | -0.158      | 2021-05-14 01:07:49 | -0.149      |
| 2021-05-14 01:07:20 | -0.158      | 2021-05-14 01:07:50 | -0.149      |
| 2021-05-14 01:07:21 | -0.158      | 2021-05-14 01:07:51 | -0.148      |
| 2021-05-14 01:07:22 | -0.158      | 2021-05-14 01:07:52 | -0.147      |

Table S1. The detection limit.

| 2021-05-14 01:07:23    | -0.158 | 2021-05-14 01:07:53 | -0.146 |  |
|------------------------|--------|---------------------|--------|--|
| 2021-05-14 01:07:24    | -0.158 | 2021-05-14 01:07:54 | -0.146 |  |
| 2021-05-14 01:07:25    | -0.158 | 2021-05-14 01:07:55 | -0.145 |  |
| 2021-05-14 01:07:26    | -0.157 | 2021-05-14 01:07:56 | -0.144 |  |
| 2021-05-14 01:07:27    | -0.157 | 2021-05-14 01:07:57 | -0.144 |  |
| Standard deviation     |        | 0.0056              |        |  |
| Detection limit (ppbv) |        | 0.017               |        |  |
|                        |        |                     |        |  |

After completing the multi-point calibration, three different concentrations of standard solutions were measured for 30 minutes under the liquid measurement mode, and the zero-gas mode was used at 10-minute intervals. Take 300 groups of data in each liquid measurement mode of the same concentration standard solution, and calculate the accuracy and repetition rate of three different concentration standard solutions respectively. Table S2 shows the accuracy and repetition rate of three different concentration rate of three different concentrations in our study.

| Table 52. The accuracy and repetition rate.      |                                   |              |                     |  |  |  |
|--------------------------------------------------|-----------------------------------|--------------|---------------------|--|--|--|
| Standard solution (ppbv / $\mu g \cdot L^{-1}$ ) | Measured ( $\mu g \cdot L^{-1}$ ) | Accuracy (%) | Repetition rate (%) |  |  |  |
| 10 / 29.892                                      | 29.683                            | -0.70        | 1.00                |  |  |  |
| 25 / 74.709                                      | 74.811                            | 0.14         | 0.06                |  |  |  |
| 40 / 119.499                                     | 123.444                           | 3.30         | 0.64                |  |  |  |

Table S2. The accuracy and repetition rate

The detailed description of the HCHO analyzer was added in the manuscript and the Supplementary, and the relevant revised content in the manuscript is as follows:

"HCHO analyzer (FMS-100, Focused Photonics Inc., Hangzhou, China) was used to monitor the HCHO mixing ratios with a temporal resolution of 1 s, which collected gaseous HCHO at a flow rate of 1 L·min-1 by an H2SO4 stripping solution and quantified HCHO mixing ratios through detection by fluorescence at  $\lambda$ =510 nm (Hu et al., 2022; Glowania et al., 2021). The HCHO liquid solution quantification is based on the Hantzsch reaction, but the reaction speed is slow at room temperature (Glowania et al., 2021). In our study, stainless steel tube for heating efficiency and ceramic fiber board thermal insulation layer for thermal insulation efficiency were used to control the reaction temperature of the mixed solution in the reaction chamber to shorten the reaction time, which reduced the time required for thermal equilibrium significantly, achieving a high signal acquisition frequency. The different dilutions of the HCHO standard solution and a blank were used to make a multi-point calibration every week for obtaining a curve with R2 ≥0.999. In these conditions, the limit of detection was 50 pptv and the uncertainty was ≤5% in this study. The detailed detection limit, accuracy, and repetition rate testing were shown in Table S1 and Table S2." Regarding interpretation of a highly-constrained model: Throughout the text, attention is given to the difference between HCHO production and loss rates (described as "net production rate" on L181). The model, however, is forced to measured HCHO. How well does the model predict HCHO if this constraint is turned off? If the model performs poorly, this calls into question the utility of the "net production rate" since the HCHO concentration does not match what would be predicted based on the modeled gross production rate.

**Response:** Thanks for your suggestion. We turned off the observed HCHO values to discuss the model predicting HCHO, and Figure S1 shows the simulated and observed HCHO at the study site. In general, the model overestimated HCHO concentration. According to previous studies, the inconsistency between simulated and observed HCHO could be caused by the uncertainties in the treatment of dry deposition, faster vertical transport, uptake of HCHO, atmospheric diffusion/dilution meteorological conditions, and fresh emission of precursor VOCs (Li et al., 2014; Zhang et al., 2021). The index of agreement (IOA) (Zhang et al., 2021), which was calculated by the differences between the modeled HCHO concentrations and observed concentrations, is used to judge the rationality of the model results (detailed introduction of IOA was shown in the first question of (Professor Ye Referee #1).

The IOA range is 0-1, and the higher the IOA value is, the better agreement between modeled and observed values is. In many studies, IOA ranges from 0.68 to 0.89 (Wang et al., 2018), and the modeled results are reasonable. The IOAs in spring and autumn in our research are 0.83 and 0.80, respectively. Although there is a certain discrepancy, the model could generally reflect the atmospheric chemical processes, and these results still provide valuable information on secondary formation of HCHO at our study site.

Figure S1. The simulated and observed HCHO at the study site.

PMF analysis: Several questions here.

1. Why are other species not included in PMF (CO, NOx, PAN)? In particular, CO should be a clear marker of vehicle exhaust.

**Response:** Thanks for your suggestion. Our idea is to analyze the source distribution of VOCs, and then determine the contribution of different sources to HCHO. Hence, we chose HCHO, 17 NMHCs, 1,2-dichloroethane, and O3 to put into the PMF model together, and these species were selected because most of them are typical tracers of specific sources and have relatively high concentrations. Among them, O3 is used as a surrogate for photochemical processes to determine the secondary fraction of HCHO, and the NMHCs of 3-methylpentane, iso-pentane, the light hydrocarbons of n/iso-pentane and n/iso-butane also are good indicators of vehicle exhaust. Hence, the species of CO, NOx, and PAN are not included in PMF. The model validation in our study indicated PMF reasonably identified the contributions of primary and secondary sources of HCHO, and the detailed model validation information shows in the first question of Professor Ye (Referee #1). In previous studies, the researchers of Ling et al. (2017) and Zeng et al. (2019) adopted the same method to analyze HCHO based on PMF. The detailed introduction is also added to the manuscript in Section 2.2, as follows:

"In our study, we chose HCHO, 17 Non-Methane Hydrocarbons (NMHCs), 1,2dichloroethane, and O3 to put into the PMF model, which are typical tracers of specific sources and have relatively high concentrations. Among them, O3 is used as a surrogate for photochemical processes to determine the secondary fraction of HCHO (Ling et al., 2017; Zeng et al., 2019)."

2. Are the authors really suggesting that the HCHO associated with isoprene is directly emitted by the ecosystem? Is there any literature evidence of that? It seems more likely that this HCHO was produced by isoprene enroute to the site. Possibly without significant ozone production (e.g. from a nearby forest).

**Response:** We are sorry for the confusion caused by the unclear description, we have revised the related contents. As you mentioned, we also think that HCHO is produced by isoprene, and isoprene is the precursor of HCHO. Meanwhile, Section 3.3.1 of HCHO in situ formation pathways also showed that isoprene is the precursor of HCHO. The revised contents in the manuscript were as follows:

"Factor 4 was characterized by a high percentage of isoprene, and the isoprene is an important precursor of HCHO (detailed discussion in Section 3.3.1). Thus, Factor 4 was designated as biogenic source, which produced HCHO from isoprene by photochemical process (Sindelarova et al., 2022; Na et al., 2004)."

3. What is the real meaning of "secondary formation"? Again, it seems likely that the HCHO from those other sources is a mix of primary and secondary. It seems more accurate to call it "Ozone associated HCHO." This is a general shortcoming of using

PMF to parse something like HCHO and it should be acknowledged and clarified.

**Response:** We strongly agree with you that photochemical processes could lead to the deviation between the primary and secondary sources of HCHO by the PMF model. Thus, apportioning HCHO sources using the PMF model should be approached with care. The relationships between the factor contributions to each species and  $K_{OH}$  value for species in Figure 5 conform to these distribution characteristics, confirming the reasonable PMF results identified as the sources of HCHO. In our study, O3 is used as a surrogate for photochemical processes to determine the secondary fraction of HCHO. Therefore, we think it is more appropriate to interpret this as follows:

"Factor 1 was characterized by a high load of O3, attributed to the intensive photochemical processes, that is, secondary formation of HCHO (Zeng et al., 2019; Ling et al., 2017; Li et al., 2010). Meanwhile, secondary HCHO measured at the study site includes in-situ photochemical production and regional transport."

Figure 5. Relationship between the factor contributions to each species and KOH values (representing chemical activities) of the species. Each square represents one species, while HCHO is represented as a square in red.

Data and Code Availability: According to FAIR standards, the observations and box model code should be publicly available without having to request them from the author.

**Response:** Observations results are provided in the manuscript, and the box model code is publicly available on the website of http://mcm.leeds.ac.uk/MCM/.

**Specific Comments**

L24: The method for determining HCHO contributions should be stated here.

**Response:** Thanks for your suggestion, we have added the method, and the revised sentence in the manuscript is as follows:

"Positive Matrix Factorization (PMF) model results showed that secondary formation made the largest contributions to HCHO (49% in spring and 46% in autumn), followed by vehicle exhaust (25% and 20%) and biogenic emission (18% and 24%) in this study."

L59 – 62: suggest deletion of this sentence.

**Response:** Thanks for your suggestion, we have deleted the sentence.

146: why is the error factor 10% instead of actual measurement accuracy?

**Response:** We are sorry for the unclear introduction, and the error factor depends on actual measurement accuracy. Because the measurement accuracies for all the species were <10%, the uncertainty of the concentrations input into the model was set as 10% based on experience. We have corrected the description as follows:

"*EF* (error factor) is set as 10% because of the <10% measurement accuracies for all the species (Ling et al., 2016; Zeng et al., 2019)."

**L166: How is the boundary layer height determined?**

**Response:** We have done sensitivity tests of the boundary layer height, and the sensitivity model running with different maximum mixing heights (1000 and 2000 m) indicated that its impacts on the modeling results (e.g. simulated HOx concentrations and OH production rate) were negligible. Hence, we determined the mixing layer height was assumed to vary from 300 m at night to 1500 m in the afternoon, and this parameter has been widely adopted in previous studies (Chen et al., 2020; Liu et al., 2022).

L173: updating constraints at hourly intervals is too coarse and likely leads to model artifacts due to step changes in photolysis and other parameters. 10–15 minute time steps are more appropriate for science-grade simulations.

**Response:** We strongly agree with your suggestions of science-grade simulations of 10-15 minute time steps. We considered the 10-minute and 1-hour time steps, and compared the simulated and observed HCHO in the two simulation scenarios (Fig R2). In general, both the two simulation scenarios overestimated HCHO concentration, while the overestimation of the simulated HCHO value in 10-minute scenario is significantly higher than that in 1-hour scenario. As we described in our introduction section, Xiamen frequently appeared O3 pollution events in spring and autumn, because the meteorological conditions were governed by weather systems such as the quasistationary front and the west pacific subtropical high, which enhanced the formation and accumulation of photochemical pollutants. Hence, the air mass in spring and autumn is not stable. Even if we try to choose relatively stagnant weather in spring and autumn in our study, the influence of meteorology still cannot be ignored. Since the model is a 0-dimensional model lacking regional transport, the simulated results will be overestimated to a certain extent. HCHO is a reactive carbonyl compound in the troposphere, if the model constraint becomes 10 minutes, the precursors are effectively replenished, leading to the accelerated production rates of HCHO and accumulated HCHO concentrations, which will naturally amplify the impacts of regional transport. Meanwhile, the primary HCHO emissions also affect the discrepancy between simulated and observed results. As I mentioned in the second question, the IOAs in 1hour scenario are 0.83 in spring and 0.80 in autumn, which were in the reasonable IOA ranges from 0.68 to 0.89, while the IOAs in 1-hour scenario are 0.45 in spring and 0.43 in autumn. Hence, the 1 hour time step in this research was more reasonably acceptable and suitable. In related previous studies, the time step of 1 hour was widely adopted to study HCHO mechanism based on OBM (Zhang et al., 2021; Yang et al., 2020; Zeng et al., 2019).